


# Remote Sensing of solar surface radiation - A reflection of concepts, applications and input data based on experience with the effective cloud albedo

Richard Müller[DWD] and Uwe Pfeifroth[DWD]

[DWD]Frankfurter Str. 135, Offenbach

**Correspondence:** Richard Müller (richard.muerller@dwd.de)

**Abstract.** Accurate solar surface irradiance data (SSI) is a prerequisite for efficient planning and operation of solar energy systems. Respective data are also essential for climate monitoring and analysis. Satellite-based SSI has grown in importance over the last few decades. However, a retrieval method is needed to relate the measured radiances at the satellite to the solar surface irradiance. In a widespread classical approach, these radiances are used directly to derive the effective cloud albedo (CAL) as basis for the estimation of the solar surface irradiance. This approach has been already introduced and discussed in the early 1980s. Various approaches are briefly discussed and analyzed, including an overview of open questions and opportunities for improvement. Special emphasis is placed on the reflection of fundamental physical laws and atmospheric measurement techniques. In addition, atmospheric input data and key applications are briefly discussed. It is concluded that the well established observational-based CAL approach is still an excellent choice for the retrieval of the cloud transmission. The coupling with Look-Up-Table based clear sky models enables the estimation of solar surface irradiance with high accuracy and homogeneity. This could explain why, despite its age, the direct CAL approach is still used by key players in energy meteorology and the climate community. For the clear sky input data it is recommended to use ECMWF forecast and reanalysis data.

## 1 Introduction

The surface solar irradiance ($SSI$) is defined as the incoming solar radiation at the surface in the 0.2–4.0 μm wavelength region on a given surface area, and is usually expressed in $W/m^2$ (Watts per square meter). SSI consists of the diffuse part (diffuse irradiance) and the direct part (direct irradiance), which is received from the direction of the sun. Diffuse radiation results from scattering by the atmosphere and is received from all directions.

Solar irradiance is a forcing quantity for heat fluxes, which causes atmospheric motion in all scales and is therefore a driving force for the circulation system (Müller, 2012b). Solar irradiance is needed for the monitoring of the Earth's radiation budget (Hollmann et al., 2006; Mueller et al., 2009; Cox et al., 2004; Darnell et al., 1992; Gupta et al., 2001, 1999; Harries et al., 2005; Harrison et al., 1990; Ramanthan and Cess, 1989), for analysis and prediction of extremes like heat waves (Träger-Chatterjee et al., 2013, 2014), climate change studies (Wild, 2009; Gilgen et al., 2009) and trend analysis (Pfeifroth et al., 2018; Pinker et al., 2005; Hinkelmann et al., 2009). Further, climate indices based on solar irradiance are of value for monitoring and seasonal prediction of specific climate impacts (Wang and Qu, 2007). As a consequence solar irradiance is a key for a





better understanding of atmospheric dynamics as well as the monitoring and analysis of climate trends and variability. Further applications include the satellite based estimation of droughts and evaporation (Dobler et al., 2011), hydrological modelling (Müller-Schmied et al., 2016), agro meteorology (Müller, 2021) as well as verification of reanalysis data and numerical weather prediction (Urbich et al., 2019, 2020; Babst et al., 2008; Träger-Chatterjee et al., 2010; Urraca et al., 2018).

An outstanding socio-economic application area is energy meteorology, which is linked to a huge commercial market as-
sociated with extensive and many-sided user demands on solar radiation data. The goal of energy meteorology is to provide specific information for efficient and sustainable use of renewables. Climatological data of solar surface irradiance are neces- sary for an efficient planning and monitoring of solar energy systems, e.g. the estimation of energy yield for Photovoltaic (PV) (Huld et al., 2012), monitoring and analysis of operational PV-systems (Hammer et al., 2003) and PV Performance Studies (Amillo et al., 2015; Huld and Amillo, 2015). Near real time data and forecasts of solar surface irradiance are needed for the
efficient integration of solar energy into the electric grid. Solar energy is fluctuating, hence forecasts are needed to optimise the levelling of conventional power plants (e.g. increase/decrease of power supply) and storage systems, to minimise the need for spontaneous trading of electricity on an in-balanced market (Hammer et al., 2003, 2015; Urbich et al., 2019) and to avoid grid overloads. Supported by the European Commission (EC) goal to aim for a secure, independent, sustainable and environmental friendly energy supply (EC1, 1996; EC2, 2013) the renewable energies contribute nowadays significantly to the energy pro-
duction. The share of PV of the total electricity sold in 2018 was 7.1 % in Italy (Jäger-Waldau, 2019) and 9.3 % 2020 Germany in 2020 (Wirth, 2021). Even in less sunny regions (e.g. Denmark, UK) the share is of about 3 %. PV electricity generation capacity accounted for 39 % of the new installed capacity in 2018 (Europe), but worldwide Asia and the Pacific region had the highest share of new installed PV power capacity in 2018. Nevertheless, also the United States had reached a cumulative PV capacity of almost 62.7 GW by the end of 2018 (Jäger-Waldau, 2019). Almost all players predict a further increase of PV. As a
consequence, the demand for accurate climatological data, near real time data and forecasts of SSI data are expected to further increase.

For all mentioned applications accurate SSI data with a large geographical coverage and a high spatial and temporal resolu- tion is either needed or of great benefit. SSI data gained from satellite observations meet these requirements and are therefore widely used in many application areas. There are typically two types of satellite systems available for the retrieval of SSI.
Geostationary weather satellites with a fixed location above the equator and polar orbiting satellites that fly in low orbit around the Earth. flying in low orbit around the Earth. Geostationary satellites provide images in high spatial and temporal resolution, but are blind near the poles. This gap can be filled by polar orbiting satellites or by data from Numerical Weather Prediction (NWP). Geostationary satellites like METEOSAT (METEOrological SATellite, Schmetz et al. (2002), GOES (Geostationary Operational Environmental Satellite GOES (2019)) or HIMAWARI (Sunflower, JMA (2017)) provide satellite data with a spa-
tial resolution of about 500m to 3 km (sub-satellite point) and a temporal resolution of about a few minutes to 15 minutes, depending on the mode of operation and spectral channel. E.g. the visible 600 nm channel of HIMAWARI has a spatial res- olution of 500 nm and a temporal resolution of 10 minutes, with even higher temporal resolutions in rapid scan mode. These satellites cover the complete geostationary ring and are quite useful for the estimation of solar irradiance at any point outside the polar region.



The respective retrieval methods for SSI are primary based on the law of energy conversation. Radiation that is not reflected or absorbed by the atmosphere will reach the ground. Clouds are the main component for the atmospheric reflection. The satellite based observation of radiances reflected by clouds is the basis for the calculation of the cloud transmission. The combination with information of the clear sky transmittance enables the retrieval of gridded radiation data sets with large geographical coverage and high spatial and temporal resolution. In many regions of the world ground based measurements are
quite rare, (e.g. African continent, oceans). A look on the station list of the global high quality network BSRN (Ohmura et al., 1998) reflects the scarcity of well maintained solar radiation measurements. Over oceans satellite data are needed for climate monitoring because of the lack of almost any reliable ground based SSI measurements (Behr et al., 2009). Thus, for most of the world satellite based solar irradiance data are the primary observational source. Moreover, for any application requiring direct irradiance, satellite data are the main observational source, also in regions with a dense network of SSI measurements. This
also applies for spectral resolved irradiance. The respective applications cover beside others the estimation of sunshine duration (WMO, 2010), the calculation of SSI on tilted planes and PV studies (Huld and Amillo, 2015; Amillo et al., 2015). Finally, satellite based SSI data are also a powerful alternative and complement in regions with a dense network of well maintained ground based pyranometer measurements, e.g. (Perez et al., 1998; Journée et al., 2012).

Satellite-based SSI data therefore play a key role in almost all applications today and also add important information in
countries with a dense ground-based network. The value of satellite data is further increased due to the automation of ground based networks. As a consequence, satellite based solar surface radiation data sets are available from different sources all over the world. Open access to data and services are essential for science and for the efficient economical use. Thus, this review focuses on the retrieval methods used to generate open data. Within this discussion a climate data record (CDR) is hereafter referred to as a time series of sufficient length, consistency, and quality to determine climate variability and change.
A CDR enables the accurate monitoring and analysis of climate trends and extremes. Essential data provider in Europe are the Satellite Applications Facilities (SAFs). SAFs are dedicated centres for processing satellite data, focusing on different application areas. They are part of the European Organisation for the Exploitation of Meteorological Satellites (EUMETSAT) and belong to the EUMETSAT network of SAFs (Schmetz et al., 2002). They are funded by EUMETSAT and the SAF consortium members, which consists mainly of meteorological services and institutes. Three of the SAFs provide satellite
based solar surface irradiance. The Climate Monitoring SAF (CM SAF), the Land Surface Analysis SAF (LSA SAF) and the Ocean and Sea Ice SAF (OSI SAF). The CM SAF generates and provides satellite based Essential Climate Variables (ECVs) related to the energy and water cycle for the analysis and monitoring of the climate system, please see (Schulz et al., 2008) for more details. Beside surface radiation also cloud parameters, top of atmosphere (TOA) radiation budget components, atmospheric water vapour and precipitation data are part of the CM SAF data portfolio (Woick et al., 2002). The methods for
the retrieval of the SSI are described in Mueller et al. (2009, 2011, 2012a); Mueller and Trentmann (2015); Müller et al. (2015) and Posselt et al. (2011b). The data can be ordered via the CM SAF web user interface (wui.cmsaf.eu). The LSA SAF develops methods for the retrieval of land surface products, such as radiation products, surface albedo, snow cover, evapotranspiration and wild fires. The respective near real time products are provided via the EUMETcast service. LSA SAF methods are also used within the COPERNICUS services. The method for the retrieval of SSI is described in Carrer et al. (2019). The OSI



SAF develops, processes and distributes near real-time products related to the ocean-atmosphere interaction. This includes, scatterometer winds, sea ice concentration, sea surface temperature, as well as radiation products, which are also generated over land. Products are accessible on OSI SAF FTP servers, EUMETCast, and EUMETSAT Data Centre (EDC). The method for the retrieval of surface radiation is described in Marsouin (2019) and follows the approach of Gautier et al. (1980).

The Global Energy and Water Exchanges (**GEWEX**) project is dedicated to understanding Earth's water cycle and energy
fluxes. GEWEX is a network of scientists. Thus, several data sets are offered free off charge under the umbrella of GEWEX. Respective data sets are usually announced and discussed in the quarterly GEWEX newsletter (GEWEX-Quarterly). The International Satellite Cloud Climatology Project (**ISCCP**) was established in 1982 as part of the World Climate Research Program (WCRP). A main goal of ISCCP is the generation of satellite based global cloud data for the monitoring and analysis of the global distribution of clouds, their properties, and their diurnal, seasonal and inter-annual variations. Within this scope
Bishop and Rossow (1991) developed a fast radiative transfer algorithm for calculating SSI, which uses the total cloud amount and cloud optical depth from the International Satellite Cloud Climatology Project (ISCCP) as important input parameters. The ISCCP method for cloud detection is described in Rossow and Garder (1993). Respective data is available from the ISCCP web page and as SSI data set from National Aeronautics and Space Administration - Goddard Institute for Space Studies (NASA-GISS). The Clouds and the Earth's Radiant Energy System (**CERES**) teams generate and provide data of the Earth
Radiation Budget. The satellite instruments used for CERES products are developed for NASA's Earth Observing System (EOS). The first CERES instrument was launched in December 1997 aboard NASA's Tropical Rainfall Measurement Mission (TRMM). CERES instruments are collecting observations on different satellite missions, including the EOS Terra and Aqua observatories, as well as the Suomi National Polar-orbiting Partnership (S-NPP) observatory of NASA and NOAA.

Solar energy specific data are provided from several sources as well. Joint Research Centre's PhotoVoltaic GeoInformation
System PVGIS (*ec.europa.eu/jrc/en/pvgis*) offers an online tool for the estimation of the expected PV yield for user selected sites. Several parameters can be chosen online, e.g. the slope of the PV module, the kind of solar cell. As an additional service PVGIS allows to visualise and download solar surface radiation (broadband) for user selected sites. Hereby different data sources can be selected, in addition to the satellite based CM SAF SARAH data sets (Mueller and Trentmann, 2015) also data from NWP models, e.g. ERA-5 (Hersbach et al., 2020). SSI data is also available at the SoDa servide. It originates from
a European project funded by the European Commission in 1999. SoDa is commercialised by Transvalor S.A. since 2009. However, SoDa still offers solar radiation data free of charge. Please see the the SODA web page (*www.soda.de*) for further information. Satellight (*www.satellight.com*) is a European data base for daylight and solar radiation. The starting point of the project was also a European project. The main data source in the US is NREL (Cox et al., 2018; NREL1)

## 2   The methods - Basics

SSI results from the emission of radiation from the sun. Without scattering and absorption by the atmosphere the SSI would only depend on the solar zenith angle and would be identical to the incoming solar radiation at the Top Of Atmosphere (TOA).





The atmosphere modifies the amount and distribution of the TOA solar irradiance by scattering and absorption. All retrieval methods rely on the same basic equations, which is based on the law of energy conservation, expressed here as

$$T + R + A = 1 \tag{1}$$

$T$ is hereby the Transmittance, which defines the proportion of radiation which passes through a medium. R is the proportion which is reflected and $A$ the proportion that is absorbed by the medium, respectively. The sum of transmitted radiation plus the absorbed radiation plus the reflected radiation is 1. This means radiation that is not absorbed or scattered back by the atmosphere reaches the surface, leading to the following basic equation, see Müller (2012b) for further details.

$$SSI = SSI_{clear} * T_{cloud} \tag{2}$$

with

$$SSI_{clear} = T_{clear} * SSI_{ext} * cos(SZA) \tag{3}$$

    SSI is the solar surface irradiance, $SSI_{clear}$ is the clear sky irradiance, $T_{cloud}$ is the cloud transmission, $SSI_{ext}$ is the extraterrestrial irradiance, often approximated as solar constant and cos(sza) is the cosine of the solar zenith angle. Various algorithms and methods have been developed to estimate broadband surface solar radiation data sets based on equation 2 e.g.,
(Möser and Raschke, 1984; Cano et al., 1986; Bishop and Rossow, 1991; Pinker and Laszlo, 1992; Pinker et al., 1995; Darnell et al., 1992; Rigollier et al., 2004; Mueller et al., 2009; Dürr and Zelenka, 2009; Posselt et al., 2011b; Müller et al., 2015). In the following the individual components of equation 3 are discussed concerning applied methods and needed input information. Please note that all quantities in the above equations are fluxes and not radiances.

### 2.1   Extraterrestrial irradiance

The extraterrestrial irradiance $SSI_{ext}$ results from the radiation emmitted by the sun and the distance between Earth and sun and is given by equation 4.

$$SSI_{ext} = SSI_{ext,const} * d_{cor} \tag{4}$$

    Here, $SSI_{ext,const}$ is the so called solar constant, a long term mean of the extraterrestrial irradiance for a mean distance between the Earth and the Sun ($d_{cor}$ = 1). Nowadays accurate values of $SSI_{ext}$ can be gained from satellite measurements
(Kato et al., 2013; Harries et al., 2005). The University Corporation for Atmospheric Research (UCAR) proposes a value of 1361 W/m$^2$ for $SSI_{ext,const}$ (UCARteam).

    $SSI_{ext}$ is valid for the whole globe and results from the nearly constant flux of energy emitted by the sun (Carroll and Ostlie, 2017). However, the term solar constant is somehow misleading, as there are variations in the solar irradiance at the top





of atmosphere due to sun activity. The changes of the solar irradiance due to variations in sun activity are usually within the
range of 2-3 $\mathrm{W/m}^2$ and are therefore often neglected (Müller, 2012b).

In addition, variations in extraterrestrial irradiance are caused by the distance between Earth and Sun. Induced by the ellip-
tical orbit of the Earth around the sun, the Earth-Sun distance varies sinusoidal throughout the year in the order of +/- 3.3 %,
leading to an respective increase and decrease of solar irradiance at the top of atmosphere. This effect is considered in equation
4 by the factor $d_{cor}$ and can be gained from standard astronomical equations (Iqbal, 1983).

## 2.2   Solar zenith angle

The Earth is approximately a sphere, hence, close to the equator the incoming solar irradiance covers a smaller region than
at higher latitudes. Here, the same amount of incoming photons are spread over a larger region, hence the solar irradiance is
lower per unit area. In mathematical terms this effect is expressed by the cosine of the solar zenith angle. The solar altitude
is defined as 90 degree minus the solar zenith angle. As can be seen in equation 3, the solar zenith angle is a dominant factor
165     for solar surface irradiance. The local noon solar zenith angle and its diurnal and seasonal variation depend on the latitude,
longitude and time of the year. These variations originate from the rotation of the Earth around it´s axis in combination with
the rotation of the Earth around the Sun on an elliptical sphere. The Earth's rotation axis is tilted at  23.5 degree with respect to
the ecliptic axis. This tilt is called the obliquity of Earth's axis and is responsible for winter and summertime. The higher the
latitude the lower the local noon solar zenith angle, which decreases from summertime to wintertime. The sun earth geometry
170     is well defined and the SZA can be calculated with standard astronomical equations (Iqbal, 1983) using latitude, longitude and
time as input. The calculation of the SZA is therefore straightforward.





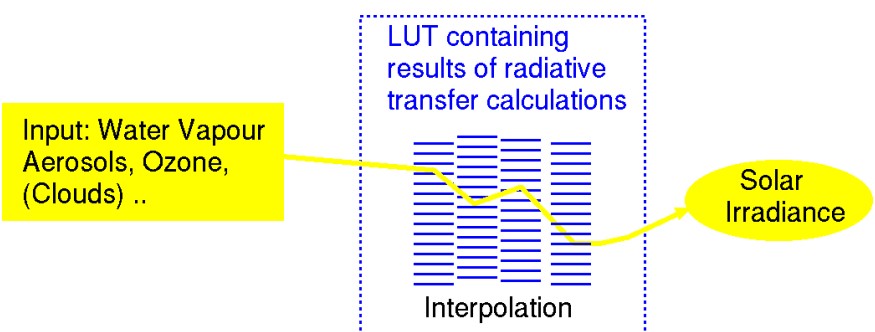

**Figure 1.** The principle of an LUT approach. The transmission is pre-calculated for a variety of atmospheric states with a radiative transfer model (RTM) and saved in a look-up table (LUT). The LUT is then used to determine SSI based on the input given for the atmospheric state for each pixel and time step by interpolation.

## 2.3 Radiative Transfer Models

The cloud and clear sky transmission needed to derive SSI (equation 2) can be calculated by a Radiative Transfer Model (RTM) with high accuracy for any given atmospheric state. With libradtran (Mayer and Kylling, 2005; Emde et al., 2016) a powerfull open software pacckage is available for RTM calculations (www.libradtran.org). Also for 3-d modelling RTMs are available in open access, e.g. Spherical Harmonic Discrete Ordinate Method (SHDOM) for Atmospheric Radiative Transfer (Pincus and Evans, 2009). An exemplary application of SHDOM within the scope of remote sensing of SSI is discussed in (Girodo, 2003; Girodo et al., 2006). However, concerning satellites an explicit usage of RTM for each pixel and time step would be too slow for the generation of long time series (climate data records) or near real time applications. Thus, instead of explicit RTM runs pre-calculated look-up-tables (LUTs) are usually used. A Look Up Table (LUT) contains the pre-calculated results of radiative transfer calculations performed for many different states of the atmosphere, surface albedo, and solar zenith angles. Figure 1 illustrates the principle of the LUT approach.

Nevertheless, classical LUT approaches have still a significant disadvantage. First, the computation speed is still hampered by the need to interpolate within large multi-dimensional arrays. Second, and more importantly, many RTMs must be performed to generate the LUTs, resulting in large and cumbersome binary matrices that raise the recalculation threshold. However recalcuation , whiis necessary from time to time to take advantage of new developments. Further, big and cumbersome LUTs increase also the chance for hidden bugs. E.g. after 4 years of development the CM SAF classical LUTs had a bug , probably induced by a switch in the indices somehow. Thus, 4 years of development were more or less worthless. The LUTs (clear sky, cloudy sky) contained over 100 000 RTM calculations each. As a consequence of the mentioned handicaps and the burden of a recalculation of classical LUTs a novel approach was developed, referred to as the "eigenvector hybrid" LUT approach (Mueller et al., 2009) . This method takes benefit of the symmetries of atmospheric absorption processes and separates them from the linear independent processes of aerosol scattering. A further reduction is achieved by the use of the Modified Lambert Function (Mueller et al., 2004; Ineichen, 2008), which requires only two SZA points (0 and 60 degree) to cover the complete





SZA dependency in good accuracy. Overall, the number of needed RTM calculation could be scaled down from 100 000 to
about 100 without loosing accuracy (Mueller et al., 2009, 2012a). The RTM calculations were performed with libRadtran
(Mayer and Kylling, 2005) using the DISORT solver (Stammes et al., 1988). The respective retrieval procedure enables also
the calculation of direct irradiance, resulting from a adaptation of the Skartveit et al. (1998) model for diffuse irradiance.
Overall, this work implements several of the requirements discussed in (Pinker et al., 1995) for the improvement of solar
surface irradiance retrieval, RTMs are not only needed for the calculation of LUTs but are also essential for the development
of parameterizations and a better understanding of the processes in the Earth atmosphere.

## 2.4 Clear sky transmittance

The clear sky irradiance can be derived by multiplication of the extraterrestrial irradiance $SSI_{ext}$ with the cosine of the solar
zenith angle and the clear sky transmittance $T_{clear}$

$$SSI_{clear} = T_{clear} * SSI_{ext} * cos(SZA) \tag{5}$$

The term clear sky radiation is commonly used, but nevertheless misleading. It is the radiation received in cloud free skies,
which can still be quite turbid (low visibilities, high aerosol load). The incoming irradiance is scattered and absorbed by gases
and particles in the atmosphere. Scattering and absorption lead not only to attenuation of light but also modify the spectra
of the extraterrestrial irradiance. Scattering leads to a continuous modification of the spectra whereas absorption occurs in
specific spectral bands, the so called absorption bands. In these bands the absorption is strong and the radiation is significantly
reduced. E.g. $O_3$ is a strong absorber in the UltraViolet (UV), hence almost all of the UV-A radiation reaching the top of
atmosphere is absorbed in the atmosphere, which is great luck for us, because it causes skin cancer. Further, water vapour has
strong absorption bands in the InfraRed (IR), but also in the microwave, thus acting as strong greenhouse gas. The absorption
bands result from the transition of radiation into periodic vibrations of molecules or atoms (IR) or change in electrons state
(UV, VIS), whereby the energy of the radiation is used to reach a higher energy level. The effect of the atmospheric scattering
is apparent every day. The blue sky results from Rayleigh scattering at air molecules (particles smaller than wavelength $\lambda$).
The air molecules scatter the incoming irradiance with a wavelength dependency of $\lambda^4$. This means that the scattering effect
decreases in the power of 4 with increasing wavelength. As a result, blue light is scattered and reflected much more by the sky
than other colours leading to a blue illumination of the sky (blue sky). In contrast, cloud droplets and aerosols are leading to
Mie scattering, which has a wavelength dependency of $\lambda^{-1}$ (particles are about the same size as wavelength and assumed to
be spheres). Thus there is only a small dependency on the wavelength leading to grey sky. Scattering forces the light to deviate
from the direct path and is induced by the interaction of the photons with the respective media as a consequence of oscillating
electric fields or by consideration of the wave-particle dualism of quantum physics as particle to particle collisions. More
details can be found in Jackson (1998). The great majority of air molecules ($N_2$, $O_2$, $CO_2$, methane, noble and inert gases) are
well mixed and uniformly distributed and do not affect the spatial and temporal distribution of solar surface irradiance. Thus,
their effect is rather static in space and time and they pose no problem for the estimation of SSI as a static transmission can
be assumed (Müller, 2012b). In contrast, ozone, water vapour and tropospheric aerosols are not well mixed. Their distribution





is driven by atmospheric dynamics. They show significant temporal and spatial gradients and patterns on different scales. Thus, the spatial and temporal variation of clear sky irradiance is dominated by these parameters. Yet, the effect of ozone on the broadband solar irradiance is rather small and is therefore neglected in some methods. Sources of information for these

variables are discussed in detail in section 7.3. Figure 2 shows the irradiance at the top of atmosphere and the surface on an inclined plane at 37 degree tilt (toward the equator) for a US standard atmospere and aerosol load and a SZA of 48.18 degree. The respective data was downloaded from NREL (NREL2, 1998), where more details on the atmsopheric components are given.

Different methods are applied to consider the effects of the clear sky components on the solar surface irradiance. A set of

models use semi-empirical functions to derive the clear sky irradiance, e.g. (Ineichen and Perez, 2002; Beyer et al., 1996; Cano et al., 1986; Rigollier et al., 2004; Bird and Hulstrom, 1981; Sengupta and Peter, 2003; Perez et al., 2002). More complex methods are based on look up tables derived from radiative transfer modelling e.g. (Gupta et al., 2001; Mueller et al., 2004, 2009) or some kind of hybrids of the two. However, as a satellite retrieval of $SSI_{clear}$ is not needed for a proper and accurate estimation of $SSI$ (Mueller et al., 2012a, 2009, 2004; Müller et al., 2015), the futher discussion of the methods focus on the retrieval of the cloud transmittance.

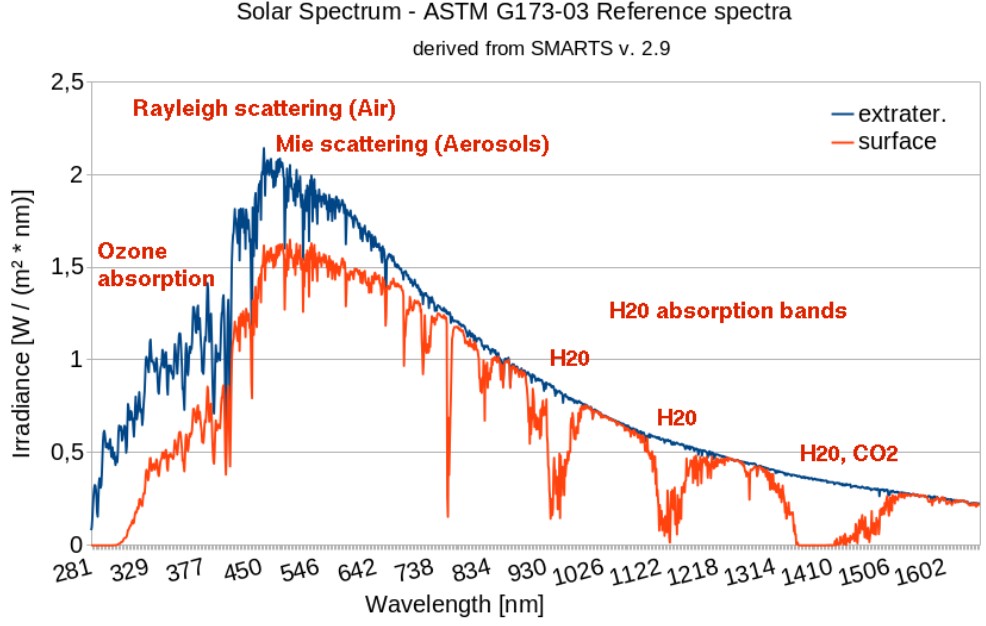

**Figure 2.** Illustration of the solar irrdiance spectrum at the top of atmosphere (extraterrestrial irradiance) and the surface for cloud free skies. The main processes leading to the attenuation of irradiance are shown in the associated spectral range.






## 2.5 Cloud transmission

Overall, clouds are the dominant atmospheric component for the spatial and temporal variation of solar surface irradiance. Thus, accurate information of the cloud transmittance is crucial for the accurate retrieval of solar surface irradiance. Satellites measure the reflection of the surface and atmosphere. In order to derive the cloud transmission the reflection of the surface

and clear sky atmosphere has to be separated from that of the clouds. The scattering effect of clouds is by far dominant and absorption can be neglected in good approximation. Thus, radiation that is not reflected by clouds is transmitted trough the clouds. Therefore cloud transmission equals in good approximation one minus cloud reflection, see equation 1.

$$T_{cloud} = 1 - R_{cloud} \qquad (6)$$

Here, $T_{cloud}$ is the cloud transmission and $R_{cloud}$ the cloud reflection. It is important to mention here that the above equation

is valid for fluxes. Satellites however, are observing radiances, i.e. photons reaching the satellite from a specific direction, see figure 3 for illustration. The reflection of the Earth's atmosphere is usually not isotropic (equal share of radiation in all directions) but depends on the specific direction and viewing angle. Therefore angular distribution models (ADM), which results from Bidirectional Reflectance Distribution Function (BRDF) have to be applied to transfer radiances to fluxes (Nicodemus et al., 1977) . See Loeb et al. (2005) for further details about BRDF and ADM.

Further, the upward flux depends on the solar zenith angle as the Surface Albedo (SAL) changes with changing SZA (Briegleb and Ramanathan, 1982). Further, the retrieval of SAL is characterised by severe uncertainties (Shuai et al., 2020; Carrer et al., 2010, 2018; Fell et al., 2021). This adds further difficulties to the accurate definition of ADMs (BRDFs) as the problem is somehow ill-posed. Is the change in the fluxes induced by a different viewing geometry for the same SAL or by the change in SAL induced by SZA, change in vegetation, different pixel size, calibration issues (ageing of channels, change of

spectral response function), change of satellite instruments, and others. BRDF and ADM are also spectrally dependent, hence differ from wavelength band to wavelength band. Thus, they can not be easily transferred from one satellite instrument to another. This poses the question which path is the best to follow, because there are mainly two different paths to get the cloud transmission. We follow the suggestion of Skartveit and Olseth to name them direct and indirect paths. For the indirect path fluxes and thus ADMs and accurate SAL information are needed. For the direct path this information is not needed but implic-

itly considered by the satellite observations within the retrieval of the cloud transmission. The two paths are briefly discussed in the following sections.

## 3 The indirect paths: Methods using flux quantities

The cloud transmission can be defined by the micro-physical parameters cloud optical depth ($COD$) and the effective radii ($r_{eff}$) of cloud droplets. Thus, one way forward is to retrieve cloud optical depth and effective radii, following an approach

discussed in Nakajima and King (1990). Once $COD$ and $r_{eff}$ are retrieved the cloud transmission can then be calculated based on RTM parameterisation or by RTM based Look Up Tables (LUTs). E.g. Deneke and Feijt (2008) use information of





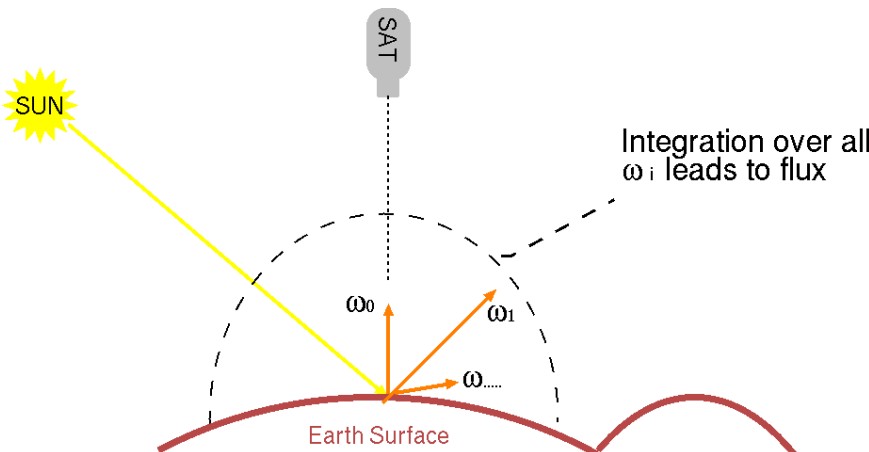

**Figure 3.** Flux is defined as the integration of all radiances over the sphere. The satellite observes the radiance for a specific direction. The reflection of the earth atmosphere is usually not isotropic (equal amount of radiation in all directions) but depends on the specific direction and viewing angle.

cloud physical properties retrieved with a LUT approach for the estimation of surface solar irradiance. The respective retrieval of $COD$ and $r_{eff}$ was developed by the Royal Netherlands Meteorological Institute (KNMI) and is applied to retrieve cloud optical thickness, liquid water path and particle size from satellite imagers within the CM-SAF framework (Roebeling et al.,

2005). A similar approach was used by ISCCP (Rossow and Zhang, 1995; Zhang et al., 1995) for the generation of CDRs. As mentioned in the method section, several approaches for the retrieval of solar surface irradiance are using the cloud optical depth and the effective radii for the estimation of the cloud transmission. However, for the retrieval of $COD$ and $r_{eff}$ accurate surface albedo (surface reflection) and ADMBRDF simulations are needed in order to separate the reflection by clouds from that of the surface and to transfer radiances to fluxes. Further, the clouds are usually assumed to be plane-parallel, which is a

rough assumption. Finally, the problem is somehow ill-posed as the retrieval of $COD$ depends on $r_{eff}$ and vise versa, and on the ratio between ice and water particles in mixed phase clouds, see Nakajima and King (1990) for details. However this ratio is usually unknown. For some of these effects a quantitative analysis is given by Kato et al. (2006)

Another indirect approach takes benefit of the linear relation between Transmission (T) and the Top of Atmosphere (TOA) albedo, as discussed in (Li et al., 1993). For a given surface albedo and clear sky state the cloud transmission changes linearly

with increasing $COD$. Thus, assuming a standard plane-parallel cloud, the relation between transmission and TOA albedo can be pre-computed for a variety of $CODs$, surface albedos and clear sky parameters. The precomputed RTM results are saved in a Look-Up-Table (LUT). The observed TOA albedo is then used as variable to retrieve the cloud transmission by interpolation within the LUT. This approach has been the basis for the first operational version referred to as prototype of the CM-SAF scheme, described and validated in Mueller et al. (2009) and Hollmann et al. (2006). This prototype followed the concepts

and ideas of Pinker and Laszlo (1992) and Pinker et al. (1995) . Also, the OSI SAF retrieval procedure for $SSI$ is using TOA





Albedo as input (Marsouin, 2019; Gautier et al., 1980). However, also in this approach clouds are assumed to be plane-parallel, ADMsBRDFs are needed to convert radiances to fluxes and accurate SAL data are necessary.

In brief, using the indirect approach means to be further away from the observations, to introduce further error sources by weak assumptions and to use simulations instead of observations directly. All in all, there are additional variables and functions
needed, which are partly based on ill-posed retrievals.

## 4   The direct path

For the indirect paths (flux based methods) accurate SAL and ADMsBRDFs are needed. Further the clouds are assumed to be plane-parallel. These induces uncertainties into the estimation of the cloud transmission and motivates the use of direct path. The direct path relates the measured reflection (radiances) directly to the cloud transmission without the need of any radiative
transfer modelling, ADM or SAL, as they are implicitly considered by observations. In the following the method applied at CM SAF and DWD is described in more detail. It is based on the concepts of (Möser, 1983; Möser and Raschke, 1984; Cano et al., 1986; Diekmann et al., 1988) and originates from the Heliosat code developed at the University of Oldenburg (Beyer et al., 1996; Hammer, 2000; Hammer et al., 2003). Only observed radiances are needed in order to retrieve the Effective Cloud Albedo (Mueller et al., 2011), which is also referred to as cloud index or effective cloud coverage. The effective cloud albedo
($CAL$) is defined as the normalised relation between the all sky and clear sky reflection derived from satellite images in the solar spectral range. $CAL$ is linearly related to the cloud transmission for $CAL$ values lower than 0.8. A clear sky model is used afterwards to calculate the solar surface irradiance based on the retrieved cloud transmission according to equation 2. This approach is still the basis for several state of the art retrieval schemes, described in more detail in (Dürr and Zelenka, 2009; Posselt et al., 2011b; Mueller et al., 2012a; Müller et al., 2015; Castelli et al., 2014; Pfeifroth et al., 2019b).
In order to derive the effective cloud albedo it is important to consider the effect of the solar zenith angle on the extraterrestrial irradiance. Furthermore, the dark offset of the instrument has to be subtracted from the observed reflections. Thus, the observed reflections are normalised by application of equation 7 in a first processing step.

$$\rho = \frac{D - D_0}{f * cos(\theta)} \tag{7}$$

Here, $D$ is the observed digital count (including the dark offset). $D_0$ is the dark offset, the instruments value in the dark, which is provided by EUMETSAT. The sun-earth distance variation is considered by the factor $f$, finally, the cosine of the solar zenith angle corrects the different illumination conditions at the top of atmosphere introduced by different solar altitudes. The resulting $\rho$ is referred to as normalised reflection. The effective cloud albedo is then derived from the normalised reflection $\rho$, the clear sky reflection $\rho_{cs}$ and the calibration value $\rho_{cal}$ by equation 8

$$CAL = \frac{\rho - \rho_{cs}}{\rho_{cal} - \rho_{cs}} \tag{8}$$

Here, $\rho$ is the observed normalised reflection for each pixel and time. $\rho_{cs}$ is the clear sky reflection, which is derived for every pixel and time slot separately. In principle this is done by applying statistics over the lowest reflections observed during



a typical time span of 3-4 weeks, e.g the average of the lowest reflections within an epsilon band, which can be defined by the user and is typically of about 10 % of $\rho_{cal}$. Also percentile functions can be used, but based on tests at DWD this is expected to lead to a lower performance. Further details on the method to derive $\rho_{cs}$ are given in Hammer (2000); Müller et al. (2015) and Mueller and Trentmann (2015). The need to calculate $\rho_{cs}$ for every time slot of the day results from the strong dependency of the clear sky albedo (which is in turn usually dominated by the surface albedo) on the solar zenith angle (Briegleb and Ramanathan, 1982). $\rho_{cal}$ is a measure for the reflections observed for the highest cloud albedos. It is therefore often also referred to the "maximum" reflection $\rho_{max}$. It is typically determined by the 95-98 percentile of all reflection values at local noon in a target region, characterised by high frequency of cloud occurrence for each month (Mueller et al., 2012a; Müller et al., 2015). In this manner changes in the reflections induced by the satellite are accounted for (Posselt et al., 2011a). Further, it calibrates also the reflections between clear sky ($CAL$=0) and "maximum" overcast ($CAL$=1).

For $CAL$ values below 0.8 the relation between $CAL$ and $T_{cloud}$ is simply given by

$$T_{Cloud} = 1 - CAL \tag{9}$$

Thus,

$$SIS = (1 - CAL) * SIS_{clear} \tag{10}$$

For very thick clouds SSI does not decrease linearly any more with $CAL$ as a result of lateral scattered light and a modification of the relation 10 is needed, e.g. as following.

$$T_{Cloud} = 1.1661 - 1.7814 * CAL + 0.7250 * CAL^2 \tag{11}$$

The term Heliosat is widely used in the energy meteorology community for methods using the relation given in 8. Yet, this naming is somehow misleading. In some Heliosat methods the observational approach for $\rho_{cs}$ is partly replaced, e.g Heliosat-2 (Rigollier et al., 2004), or completely replaced, e.g. Heliosat-5 (Tournadre et al., 2021), by modelling of $\rho_{cs}$. However, the planetary reflection is in good approximation observed by the satellite and a seperation of the effect of ground albedo and atmospheric reflection or a complete replacement of $\rho_{cs}$ with RTM simulations is a detour adding additional uncertainties. The motivation for these modifications is incomprehensible both from the point of view of measurement technology as well as from the point of view of radiative transfer modelling (see section 3 and the open reviewer comments concerning Tournadre et al. (2021)). In order to avoid misunderstandings we introduce the term CALSAT for the described method and use it instead of Heliosat throughout the manuscript. The CALSAT approach is applied routinely in real time at the University of Oldenburg since 1995. It was used to establish the server Satel-Light, which delivers valuable information on daylight in buildings to architects and other stakeholders (Fontoynont et al., 1997). The method was also used to provide the data for successful projects like PVSAT and PVSAT-2 (Lorenz et al., 2004; Hammer et al., 2003) which went into 24/7 operational services. All CM SAF Climate Data Records (CDRs) (Posselt et al., 2011b; Müller et al., 2015; Pfeifroth et al., 2019b) has been retrieved with the CALSAT approach coupled with a clear sky model based on the eigenvector hybrid LUT method (Mueller et al., 2009). The





respective data sets belong to the most used and validated data sets in the world covering a wide field of applications, as can be seen from recent publications (Wild et al., 2021; Kulesza, 2021; Alexandri et al., 2021; Gardner et al., 2021b, a; Farahat et al., 2021; Yang and Gueymard, 2021; Kulesza, 2021; Kulesza and Bojanowski, 2021; Fountoulakis et al., 2021; Drücke et al., 2021; Daggash and Mac Dowell, 2021). $CAL$ based SSI data are characterised by a high accuracy and homogeneity, which is a reason for the popularity, even up to quality control of ground based stations (Urraca et al., 2017, 2020). The CM SAF SSI data are regurarly validated by the CM SAF team e.g. Pfeifroth et al. (2019) but also by the user community, e.g. (Wang et al., 2018; Amillo et al., 2018; Urraca et al., 2007; Trolliet et al., 2018; Riihelä et al., 2015; Posselt et al., 2011b; Ineichen et al., 2009) and references therein. The complete list of peer-reviewed articles dealing with CM SAF radiation data is available on *cmsaf.eu* under the menu item *Publications* (CMSAFpubl). Even if only the CM SAF data is counted there are already thousands of users all over the world (Selbach and Thies, 2021). There are of course much more if the other data provider are accounted for. All CM SAF Validation Reports, Product User Manuals, Algorithm Theoretical Baseline Documents and Operations Reports are available on *cmsaf.eu* unter the menu item *Documentation*. The CAL based CM SAF SSI data went also into the PVGIS service (Huld et al., 2012; Amillo et al., 2014), which has several hundreds of accesses per week (T.Huld, personal communication). Finally, the CALSAT method is also used at DWD to generate 24/7 operational near real time data and forecasts, which are available at opendata link. In summary, the direct approach using the observable $CAL$ to define the cloud transmission ist quite well established and is used and for the creation of radiation data sets that are widely used around the world, e.g. Selbach and Thies (2021). It enables the retrieval of monthly means with an accuracy of **5.2** $\mathrm{W/m^2}$ and **7.7** $\mathrm{W/m^2}$ for $SSI$ and direct irradiance, respectively (Pfeifroth et al., 2019). Hereby, the mean absolute difference is used as metric for the accuracy. It is defined as the average of the absolute values of the differences between the ground based measurements and the satellite based irradiance first over all months and then over all stations. For the validation all available BSRN stations were used.

$$MAD = 1/n \sum_{i}^{1-n} |(I_{sat,i} - I_{ground,i})| \tag{12}$$

The uncertainty in CALSAT SSI trends is with -0.8±0.4 $\mathrm{W/m^2/decades}$ quite small, which shows the adequate homogeneity of the data set and its ability to identify climate trends and decadal variability.

Already the early CALSAT methods had significant impact. e.g. the method developed by (Möser and Raschke, 1984; Diekmann et al., 1988) was routinely applied at the German Weather Service till it was replaced by a modified SPECMAGIC (Mueller et al., 2012a) version in 2018 A similar method was also used for the GEWEX Surface Radiation Budget Project as a quality control algorithm (Gupta et al., 2001; Darnell et al., 1992). This algorithm known as Staylor algorithm was chosen by the World Climate Research Program (WCRP) SRB project for generation of surface solar irradiance for a test period of several years (Whitlock et al., 1995) and has been used to produce long term data set.

# 5   Spectral resolved irradiance

The demand for spectrally resolved irradiance has increased significantly in recent years due to new solar cell technologies and increased awareness of climate change. The importance of the spectrally resolved irradiance (SRI) for the efficient planning of





PV systems is discussed in Huld and Amillo (2015). Further, SRI is of benefit for a more extensive monitoring and analysis of the climate system and enables the generation of essential meteorological variables for several application fields, e.g.

- Daylight, which is of importance for the design and planning of office buildings and health studies, e.g., concerning Seasonal Affective Disorder.

- Ultraviolet (UV)-A and UV-B, important for weather warnings concerning the UV dose

- Photosynthetic Active Radiation (PAR), important for agro-meteorology.

However, retrieval algorithms for the generation of accurate long term series of spectral resolved irradiance covering the complete solar spectrum were not available or applied to generate CDRs until 2012. The development of SPECMAGIC (Mueller et al., 2012a), a fast and accurate method for the retrieval of spectral resolved irradiance, was a game changer. It has been used to improve the quality of the broadband radiation of the CM SAF CDRs, but also to provide the first spectral resolved solar

irradiance climate data record.

The SPECMAGIC method is a adaption of the concept for the broadband radiation (Mueller et al., 2009) to Kato bands (Kato et al., 1999) using the correlated-k approach offered as part of libradtran (Mayer and Kylling, 2005). The correlated-k method is developed to compute the spectral transmittance (hence the spectral fluxes) based on grouping of gaseous absorption coefficients. However, in contrast to Mueller et al. (2009) the cloud transmission is derived with the CAL approach in

SPECMAGIC (Mueller et al., 2012a) and only the clear sky irradiance is calculated with an eigenvector LUT. The CALSAT method was developed for broadband radiation. Thus, a spectral correction of the broadband transmission was needed, which was done by a RTM based spectral correction factor for the broadband CAL. A detailed description of the methods is given in Mueller et al. (2012a). Further validations were presented in Amillo et al. (2015). As ground measurements of spectral resolved irradiance are extremely rare, satellite derived information constitutes the primary data source for spectral resolved irradiance

within the scope of climate monitoring and solar energy applications. The validation of the spectral resolved irradiance climate data record (SRI) in Ispra (Italy) shows an accuracy of the monthly mean values of spectral resolved data higher than 0.03 $W/m^2/nm$ for wavelengths below 1000 nm and even better for larger wavelengths (Pfeifroth et al., 2019).





## 6    Forecasting and seamless prediction

Over recent decades the share of solar energy has increased enormously around the world, as briefly discussed in section 1. In
contrast to fossil and atomic energy, solar resources are fluctuating and the energy yield depends on the weather. Variations in
solar radiation, and thus in solar energy, have to be balanced, either by increase or decrease of the production from conventional
power plants or by trading energy on the electricity market. The shorter the time for this compensation, the higher the costs
and the lower the efficiency of renewables. In addition, wrong forecasts of extreme weather events might also lead to grid
instabilities and regional overloads. Accurate forecasts are therefore needed to enable an efficient, planing and operation of
the energy production and to optimise the use of renewable energies and the reduction of CO2 emissions. Beside forecasts,
instantaneous near real time data are also essential for the optimisations and monitoring of solar energy systems. As a result,
the demand for accurate near real time data and forecasts of solar surface irradiance has steadily increased. Satellite based
cloud information is well suited for the estimation of near real time radiation data and the basis for short term forecasts, which
outperforms NWP in the first hours (Lorenz et al., 2017; Urbich et al., 2019).

The satellite based short term forecast of SSI, also referred to as nowcasting, is usually done by retrieval of atmospheric mo-
tion vectors from satellite images and subsequent application to extrapolate the observed clouds into the future. The predicted
cloudiness can then used to calculate SSI with one of the methods discussed in section 4. In order to get the forecasts in the
needed spatial and temporal resolution a dense field of cloud motion vectors is needed. The use of cloud motion vectors for
nowcasting is widespread and many approaches have been proposed so far e.g. Gallucci et al. (2018) and Sirch et al. (2017)
used cloud motion vectors from SEVIRI (Spinning Enhanced Visible and Infrared Imager) on board of MSG to forecast solar
surface radiation for up to 2 hours. Further methods to gain AMVs are discussed for example in Raza et al. (2016); Wolff et al.
(2016); Antonanzas et al. (2016) and Barbieri et al. (2017). In recent years, however, computer vision techniques (Szeliski,
2011) have found their way into meteorology and have offered the advantage of a strong developer community, which is typ-
ical for open software. This led to the use of optical flow methods for the estimation of atmospheric motion vectors. Optical
flow has been used pre-dominantly for image pattern recognition in the fields of traffic, locomotion and face recognition (Sonka
et al., 2014). To our knowledge, one of the first applications in meteorology has been the utilisation of the optical flow for radar
images as described by Peura and Hohti (2004). At DWD the TV-L1 method (Zach et al., 2007) was implemented in 2018 for
the short-term forecast of radar reflectivity by Manuell Werner. This has motivated the adaptation of TV-L1 to $CAL$ images
(Urbich et al., 2018) in order to track the cloud movements and to extrapolate the CAL values with optical flow. In order to
met the requirements of the operational near real time retrieval of $CAL$ based on SPECMAGIC (Mueller et al., 2012a) has
been modified. The modified version is referred to as SPECMAGIC_NOW. The Visible Channel in the 600 nm band is used
instead of broadband visible channel, which is gained from a combination of VIS006 and VIS008 (Cros et al., 2006). Further, a
running backward statistics are performed to calculate $\rho_{cs}$. In contrast to SPECMAGIC, only 20 days are used for the statistics.
Also the estimation of $\rho_{cal}$ has been modified. Please see section 7 for further details. The near real time data and forecasts up
to +18 hours are available at the DWD open data server (opendata link).



The combination of the effective cloud albedo with TV-L 1 and SPECMAGIC_NOW delivers a novel powerful method for the short-term forecast of solar surface irradiance, as discussed in Urbich et al. (2019). Yet, for the satellite based nowcasting 2 subsequent images are needed. Before sunrise no former image in the visible is available for the estimation of the atmospheric motion vectors. Thus, the University of Oldenburg extended the Heliosat method to Infrared Images Hammer et al. (2015). For

longer forecast horizons a blending with NWP models is needed (Urbich et al., 2020; Lorenz et al., 2017).

However, forecasting SSI is not only needed within the scope of solar energy, but is also useful for weather warnings concerning UV radiation, heat waves, droughts and evaporation.

## 7  Recent developments and discussion

### 7.1  Method

Copernicus "hoping to return the science to a less cumbersome, more elegant view of the Universe suggested a heliocentric (Sun-centred) model of planetary motion" (Carroll and Ostlie, 2017) instead of the geocentric system, which was the prevailing view until then. His approach led to a much simpler description between planets and stars.

This paradigm should still be a central key for natural science. Make things as easy as possible and as complex as needed. It is always a goal of science to improve methods. However, established ways should not be forgotten by this legitimate goal. It

should not be expected a priori that more complex and more cumbersome methods lead to a higher accuracy. Science demands that there is a proof of improvement and not only a postulation based on the complexity of a method. Further, a central issue of science should be lessons learnt. Research and development should focus on the open issues to improve the methods and not trying to re-invent the wheel by ignoring existing knowledge. These arguments call for a reflection of lessons learnt within the Heliosat-3 project. The goal of the Heliosat-3 project (Betcke et al., 2006) was to supply high-quality solar radiation data

gained from the exploitation of existing Earth observation technologies by taking advantage of the enhanced capabilities of the new Meteosat Second Generation Satellites. Therefore one aim was to develop a method to replace the CALSAT method by a method using cloud micro-physical parameters. During the Heliosat-3 project, the meteorologists Skartveit and Olseth reminded us not to forget the advantages of the the direct path, using $CAL$ derived from satellite measurements. Lessons learned within Helisoat-3, $CAL$ is still used and was not replaced by an approach using $COD$ and $r_{eff}$ at the University of Olden-

burg, the leading entity of the project. Further, within CM SAF the LUT approach used in the early phase for the generation of the first data sets (Mueller et al., 2009) has been replaced by the CALSAT approach concerning cloud transmission for the generation of CDRs (Posselt et al., 2011b; Müller et al., 2015) and the generation of near real time data and forecasts (Urbich et al., 2018). For clear sky cases RTM based LUTs are still used in the CALSAT approach, beside other arguments, in order to overcome the limitation of turbidity as input (Mueller et al., 2004) .

Thus, the CALSAT approach has been not replaced by flux based methods at leading institutes and is to the knowledge of the authors still the best way forward for the retrieval of cloud transmission. The approach results from the law of energy conversation and relates the observed reflection directly to the effective cloud albedo and the cloud transmission without the need for external SAL data or ADM simulations. In this terms the CALSAT method takes full benefit of the satellite





measurements in the visible (Mueller et al., 2011). Both, the data sets of the University of Oldenburg, CM SAF and other

entities using $CAL$ are among the most validated data sets of the world. The validation results around the SAF radiation data

show that the improved CALSAT methods are in the meanwhile able to derive accurate SSI data close to the accuracy of well

maintained ground based stations (Posselt et al., 2011b; Müller et al., 2015; Pfeifroth et al., 2019). The quality is such good that

CAL based SSI data can be used for monitoring the quality of ground based measurements (Urraca et al., 2017, 2020) and for

climate trend analysis (Pfeifroth et al., 2018). The success and impact of the CALSAT approach is impressive. Nevertheless,

further improvements of CALSAT have been performed and are still under progress. Yet, many of these improvements are not

CALSAT specific, but represent general issues of satellite based retrieval of SSI. The following sections provide a discussion

of limitations and possible improvements that are to the best knowledge of the authors.

### 7.1.1    Mountains

The pixel size of the visible MSG full disk channels (VIS006 and VIS008) is with approximately 3-5 km over the European

mountains relative coarse. This leads to pixels which covers mountains and valleys and are therefore no longer spatially repre-

sentative of the area covered. The terrain is too heterogeneous and the spatial representatives is largely limited by the resolution

of the VIS006 and VIS008 channels. However, over Europe the High Resolution Visible (HRV) channel is available. It has a

sub-satellite resolution of 1x1km, which means of about 1x2 km over the European mountains. The spatial representativity is

significantly improved by the higher spatial resolution and is a central key for the improvement of the precision and accuracy

of SSI in mountainous regions (Dürr et al., 2010; Dürr and Zelenka, 2009).

Deep valleys are shadowed by mountains reducing the amount of available solar surface radiation. The shadowing informa-

tion needs to be corrected by the use of additional geographical information and geometrical functions. For this correction the

location, size and height of the mountains are needed, which are provided by a digitally elevation model (DEM). Respective

data are available free of charge from various web sites, see (OpenDEM-link). Geometric functions can be used to calculate the

solar zenith angle and azimuth angle for every date and hour, enabling to correct the shadowing based on the DEM informa-

tion. This correction is independent of the retrieval method for SSI. A respective correction as well as further mountain specific

features has been implemented into ths CALSAT approach by Dürr and Zelenka (2009). The work of Dürr and Zelenka (2009)

has been the basis for the Heliomont algorithm (Stöckli, 2013; Castelli et al., 2014), which comes with further improvements.

Further, snow covered surfaces occur with much higher frequency in mountains than in low lands. Thus, also the treatment

of clouds over snow needs more attention in the mountains. Dürr and Zelenka (2009) implemented a correction for the CAL

retrieval over snow and demonstrated that it leads to higher accuracy.

### 7.1.2    Long Lasting clouds

All methods need cloud free pixel in order to get actual information of the surface albedo (indirect flux based methods) or the

clear sky reflection $\rho_{cs}$. Outdated information of SAL or clear sky reflection can lead to significant errors in SSI. The clear

sky reflection and the surface albedo can only be derived from satellites in cloud free situations. Thus, long lasting clouds are

a serious error source for the satellite based retrieval of SSI. This is valid for all retrieval methods. Long lasting clouds can





contaminate the observation of $\rho_{cs}$. The minimisation of $\rho$ leads only to the clear sky reflection if at least one pixel is cloud free. However, especially in winter at higher latitudes (e.g. northwestern Europe) it can happen that there is no cloudless sky for 20-30 days, which leads to cloud contamination and an overestimation of $rho_{cs}$ and, in turn, to an underestimation of 510   $CAL$.

An option to reduce the uncertainties induced by long lasting cloud might be the use of simulated of $\rho_{cs}$ based on RTMs or climatological values for $\rho_{cs}$ as a backup. The former option has been implemented in SPECMAGIC_NOW, a SPECMAGIC modification dedicated and used for the 24/7 production of near real time data at DWD. However, the method for the decision when the backup is used is still in development and needs further evaluation. Another option is to fit the diurnal variation 515   of $\rho_{cs}$ and to replace outliers by the fitting function, see Heliomont for details (Castelli et al., 2014). This approach requires cloudfree skies for neighbouring slots in time and thus can only mitigate but not resolve the problem of long lasting clouds. However, it has also to be considered that long lasting clouds occur typically during wintertime at higher latitude and are therefore associated with low absolute SSI values and hence also low absolute errors. Even with a bakup for $\rho_{cs}$, CAL would be still derived directly from observations for the vast majority of cases.

### 7.1.3   Improvement of clear sky reflection, $\rho_{cs}$ method

Usually the broadband visible channels are used for the CALSAT method, either directly or as a combination of the visible channels in the spectral bands 600 nm (VIS006) and 800 nm (VIS008) according to Cros et al. (2006). However, the precision of the $\rho_{cs}$ detection might be improved by using the VIS006 channel as the reflection in this band is rather dark for vegetation. This enhances the contrast between clear sky and cloudy reflection and enables a higher precision for optical thin clouds. A 525   comparison of using different VIS channels was performed by Posselt et al. (2011a). This study showed the potential of using the VIS006 channel and motivated its use for SPECMAGIC_NOW. Experiments to improve the accuracy by using percentiles instead of the original statistical method (Hammer, 2000) did not lead to an improvement. Hence, the same method is used for the calculation of $\rho_{cs}$ in SPECMACIC (Mueller et al., 2012a) and SPECMAGIC_NOW.

Miss-classification of snow covered surfaces as clouds or clouds as snow is a serious error source for all methods. This is 530   also valid for the CALSAT method, but with some specificity resulting from the $\rho_{cs}$ method. To illustrate, let there be a certain number of snow free and snow covered days within the 20-30 day period used for the retrieval of $\rho_{cs}$, As $\rho_{cs}$ is calculated with a minimisation function $\rho_{cs}$ will capture always the snow free situation. For the snow covered days the snow will be treated as a thick cloud resulting in a large underestimation of SSI. This effect can be reduced if $\rho_{cs}$ is calculated with a Fuzzy logic approach, discussed in detail in Posselt et al. (2011b). However, as experienced by the CM SAF the Fuzzy logic approach 535   induced other artefacts, e.g. wrong $\rho_{cs}$ values in the desert. There might be another option to treat this problem. An accurate snow map can be used to flag snow covered regions and to correct $\rho_{cs}$ by simulated values if $\rho_{cs}$ shows too low values for snow. Of course this approach would be a rough workaround which needs further optimisation and adjustments based on validation studies. A world wide reference for snow cover maps are provided by NIC NOAA, referred to as the IMS snow ice map (Helfrich et al., 2018). This map is based on several satellite information including microwave and ground based information





and therefore not so prone to long lasting clouds. However, still melting periods of snow are sometimes not captured by IMS as a result of long lasting clouds as discussed in the snow workshop (Helmert et al., 2018).

In SPECMAGIC_NOW the duration used for the slot-wise detection of $\rho_{cs}$ has been reduced from 30 to 20 days. This is expected to reduce the effect of snow miss-classification as cloud, but might be a handicap for long lasting clouds. However, for Germany the operational comparison with other data sets (DWD in-situ network, SPECMAGIC) have not shown any significant
drawbacks concerning long lasting clouds so far.

### 7.2  Calibration

An accurate calibration is of very high importance for the generation of homogeneous climate data records. Thus, it is the basis to enable appropriate trend analysis. The focus of the first geostationary Meteosat satellites was the support of weather monitoring and forecasting. As a result calibration issues of the visible channels were not considered (thoroughly) for MVIRI
onboard of MFG and for SEVIRI onboard of MSG, respectively. Both instrument generations are not equiped with onboard calibration units for the visible channels. The increased awareness of climate change and the need for improved climate monitoring and analysis led also to the demand to use Meteosat satellite data for climate monitoring. As a result, the CM SAF was implemented by EUMETSAT in 1999 and the activities concerning calibration of the visible channels increased. For example, Govaerts et al. (2004) developed a method for the calibration of the SEVIRI visible channels using desert and cloud
targets as radiation references. However, longer time series are required for a CDR, which requires the use of MVIRI to extend the time series, but the method of Govaerts et al. (2004) is not applicable to MVIRI. Unfortunately the MVIRI time series is quite in-homogeneous as in the early days of MFG changes in the gauge were not unusual. Further, up to Meteosat-6 no inter-calibration between the satellite instruments was done, adding additional breaks in the observed reflections, see figure 4.

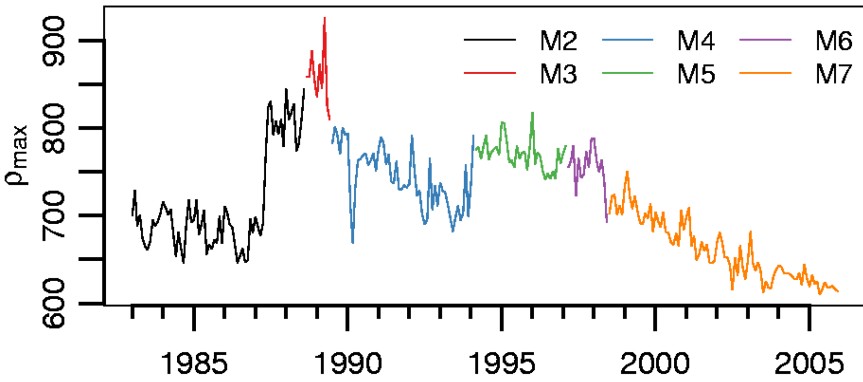

**Figure 4.** Temporal evolution of the 95th percentile ($\rho_{max}$) of the observed reflections for MVIRI, covering 1983 to 2005. The different MVIRI instruments (M 2-7) are coloured for a better distinction. Image from Posselt et al. 2011

For these reasons CM SAF took benefit of the calibration value $\rho_{cal}$ of the CALSAT method (see equation 8) for the self-
calibration of MVIRI. This is described in more detail in Posselt et al. (2011a). Basically, the $\rho$ values are ordered with a





function in a first step. Then the 95 or 98 percentile is used as $\rho_{cal}$. This is done for a target region with a high cloud coverage (Atlantic ocean, South West of Southern Africa). The advantage of this approach is that the calibration is done for the same targets (clouds) for which the transmission is retrieved. It has been shown that the method leads to stable homogeneous data sets across satellite generations (Posselt et al., 2011b; Pfeifroth et al., 2018).

This approach has been slightly modified for SPECMAGIC_NOW. First of all another target region is used, following an approach of the Royal Meteorological Insitute of Belgium (RMIB). The novel target region covers the ITC region. It is defined from line 1234 to 2486 of the Meteosat level 1.5 images. Here all counts of the 12 UTC slots are sampled over a period of 3 days and sorted with the heapsort function and subsequently assigned to a vector (fper). Then, the index of the 95 percentile (pm) and the 99.99 percentile (pmm) are calculated. Afterwards, $\rho_{cal}$ is calculated within a loop between the indices of the

respective radiances and finally multiplied by 1.1. Please see the following C-code lines for details.

```
heapsort(kend1,fper);
int pm=(int)(0.95*(kend1+1));
int pmm=(int)(0.9999*(kend1+1));
    for (kkk=pm; kkk < pmm; kkk += 1){
        Rmaxsum += fper[kkk];
        Rnum += 1;
    }
rho_cal=1.1*Rmaxsum/Rnum;
```

Comparison with the former method of CM SAF for the calculation of $\rho_{cal}$ indicates that this method leads to less noise. As

SPECMAGIC_NOW is operated 24/7 using SEVIRI as input the method of Govaerts et al. (2004) could be applied. However, concerning physics the $\rho_{cal}$ method described above has a significant advantage. Identical channels and targets (clouds) are used for both the calibration and the retrieval of $CAL$. Yet, the Govaerts et al. (2004) method uses desert targets in addition, which is expected to add uncertainties to the calibration because the spectral albedo is quite different to that of clouds. At least during the development and testing of SPECMAGIC the $\rho_{max}$ calculation could not be improved by the use of additional

desert targets.

### 7.2.1 Parallax correction

The geolocation of the satellite is always related to the Earth surface (sea level). This means that independent on cloud height the cloud is assumed to be at the surface. The higher the clouds and latitudes the larger is the displacement of the geolocation. E.g. for a cloud with a top height of about 10 km the displacement effect at mid latitudes is several kilometres. This hampers

the comparison between satellite data and ground based information, in particular if instantaneous data and not averages are compared. The parallax effect is illustrated in Figure 5. Eumetsat offers a geometrical function for the correction of parallax effect, which needs only latitude, longitude and the cloud top height (CTH) as input. The CTH can be estimated from the Brightness Temperature of the window channel at 10.8 nm. The displacement is then calculated based on geometrical (sin cos)





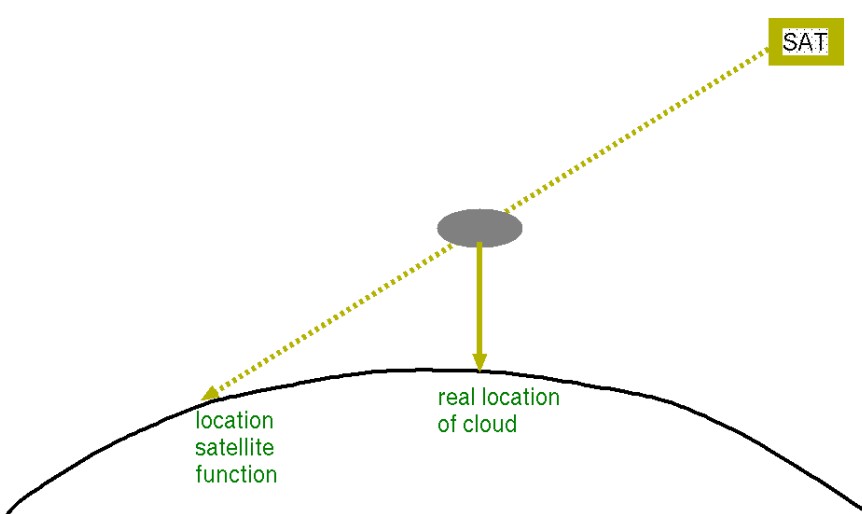

**Figure 5.** Illustration of the parallax effect.

functions. This approach works well for the Infrared channels. In the IR the satellite receives mainly the emission from the
cloud top if the clouds are not semi-transparent, which can be assumed for a COD larger than 10. Thus, CTH can be used for
the needed altitude information for the parallax correction. However, the diffuse part of $SSI$ is reflected throughout the cloud
and the height needed for the parallax correction can not be assumed to be close to the top of the cloud. Therefore, the parallax
correction should only applied to the direct radiation. The application of a parallax correction for direct irradiance was already
postulated in 1996 by Beyer et al. (1996). A parallax correction has been implemented in the HelioMont method (Stöckli,
2013; Castelli et al., 2014), but the reduction in the error measures was marginal compared to other more dominant effects
(Reto Stöckli, personal communication). One reason might be that for $COD$ larger than about 10 direct irradiance is zero. This
means that diffuse radiation usually dominates $SSI$ in cloudy situations and CTH is not well suited for the correction. Thus,
the optimal definition of the cloud height for the parallax correction needs further investigations, in particular empirical studies
and RTM calculations for different cloud scenes. After all, the parallax effect is not a weakness of a specific retrieval method,
but results from the viewing geometry. Close to the equator, everything is fine.

### 7.2.2   Slant column correction

The geostationary orbit leads to slant column viewing geometries at higher latitudes. As a consequence the "cross section" of
the clouds increases significantly, see Figure 6 for illustration. Thus, on average the cloud coverage is artificially enhanced
at larger satellite zenith angles leading to an underestimation of surface solar radiation. The validation study of Riihelä et al.
(2015) provided a clear indication for the underestimation with increasing latitude. Based on that results a brief empirical
correction has been developed and implemented in SPECMAGIC for the generation of the SARAH 2.1 data set (Pfeifroth





et al., 2019c). The correction term for the slant geometry effect is approximated as a function of the satellite zenith angle $\theta_{sat}$:

$$Corr = 0.1 \cdot ((cos(\theta_{sat}/1.13)^{1.3})^{-0.9} - 1) \tag{13}$$

The correction term is afterwards applied for $CAL * \theta_{sat}/1.3 < 0.55$ and $CAL > 0.04$

$$CAL = CAL * (1 - Corr) \tag{14}$$

This results in an improved performance of the retrieval, especially in higher latitudes (Pfeifroth et al., 2019). Yet, further empirical studies and RTM calculations are needed to improve the correction of the slant geometry effect. Again this effect is relevant for all methods. The cloud cross section increases also with increasing SZA, but here also the reflection increases as cloud optical thickness is increasing, but the slant viewing geometry of the satellite leads to an artificially increased cloud cover that does not match the ground observations.

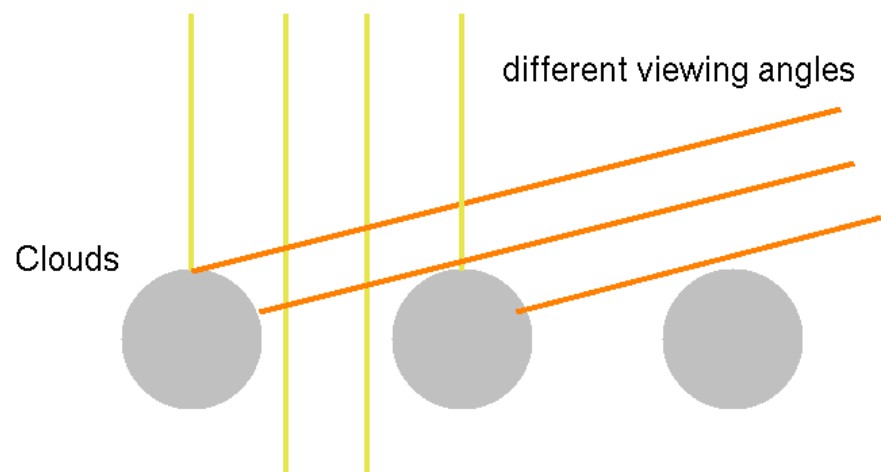

**Figure 6.** Illustration of the slant geometry effect. For a satellite zenith angle of 0 the cloud coverage would be 0.5, but for the slant geometry it would be 1.


### 7.2.3 Deep learning - neural networks

Overall, the use of artificial intelligence methods, in particular deep learning (machine learning) with neural networks, is also increasing in meteorology. This includes also the estimation of solar surface irradiance from satellites (Senkal, 2009; Yeom et al., 2019; Cornejo-Bueno et al., 2019; Dewitte et al., 2021).

However, a drawback of neural networks is the black box character and the need for retraining for new regions and satellite systems. A neural network is, strictly speaking, only valid for the training framework, as only the behaviour of the training data sets can be reproduced. An application to other regions, periods or satellite instruments typically requires extensive re-training. That is a reason why scientists started to develop spectral transfer functions to ease the adaption of the network to other satellite





instruments. However, this means using physical approaches to avoid the need for extensive retraining. Further, the black box
character hampers a deeper understanding of the involved physics and consequently the analysis of error sources. Ultimately,
it is a complex regression and the term intelligence is somehow misleading. For the retrieval of solar surface irradiance the
physics is well defined. Thus, it is not obvious what advantage NN could provide compared to physical approaches. However,
a good application of neural networks (NN) might be the replacement of LUTs. This has be done for the retrieval of SIS (e.g.
Takenaka et al. (2011)). In this study an algorithm for estimating solar radiation from space using a neural network (NN) to
approximate radiative transfer calculations were developed. Similar approaches are also used for other atmospheric variables,
e.g. for volcanic ash (Bugliaro et al., 2021). However, with respect to SSI, the burden of cumbersome LUTs, which require
hundreds of thousands of RTM calculations, is eliminated by using the hybrid eigenvector LUT concept (Mueller et al., 2009).
These LUTs are relatively easy to calculate and to manage. This might reduce the value of NN as replacement of LUTs within
the scope of SSI. Thus, the authors believe that there is no urgent need to apply neural networks for the retrieval of solar surface
radiation,in particular if the CAL approach is used for the retrieval of the cloud transmission.

### 7.3 The clear sky input data

3 SAFs are engaged in the retrieval of SSI. EUMETSAT triggered an intensive discussion and interaction between the SAFs
in order to exploit synergies. As a consequence, a comparison of the SAF SSI data (Ineichen et al., 2009) was done and
several SAF radiation workshops took place. It was discussed that all SAFs use established methods for the retrieval of the
cloud transmission, but that there is a need for action on the clear sky input data. Therefore, the clear sky input data is briefly
discussed in the following paragraphs.

#### 7.3.1 Ozone:

$O_3$ is a strong absorber and can be well detected from satellites. Data in good quality can be gained from from GOME /
SCIAMACHY (Burrows and Chance, 1992) , TOMS (Fleig et al., 1986) and the Ozone Measuring Instrument (Ziemke et al.,
2011) and the respective follow-up satellites. There is no need to use data in high temporal resolution as the spatial and
temproal variation of $SSI$ is dominated by clouds in the meso-scale. Thus, observations based on polar orbiting satellites
are not a handicap. Alternatively ozone data from the ECMWF models can be used (Bechtold et al., 2008). However, also
climatologies can be used in good approximation for broadband SSI as a deviation of about 150 DU from the US standard
atmosphere leads only to changes in SSI of about +/-5.5 $W/m^2$. This is in the range of the uncertainty of solar irradiance
measurements (Müller, 2012b).

#### 7.3.2 Water vapour:

Water vapour ($H2O_g$) is a strong absorber and the total column can be well retrieved from satellites. However, $H20_g$ can only
be retrieved in cloud free skies wihout appropriate profile information. Thus, using $H20_g$ from satellite is expected to induce
a clear sky bias in $H20_g$. Therefore, data from NWP might be the better option as the water vapour channels are assimilated



by modern state of the art NWP models (4d-VAR). Thus, this data source is strongly recommended for $H20_g$, e.g. CM SAF is using water vapour information from the ECMWF European Medium-Range Weather Forecasts NWP model (Dee and Uppala, 2009; Betts et al., 2003). Also OSI SAF uses data from ECMWF. The use of climatologies can lead to significant deviations for daily and hourly SSI (broadband) values and is not recommended as a primary choice.

### 7.3.3 Surface Albedo

Satellites are a good source for gridded Surface ALbedo (SAL) data. The optimum accuracy for the retrieval of SAL is of about 5 % (Fell et al., 2021; Riihelä and Kallio-Myers, 2020). The accuracy is limited by uncertainties in ADMs, spectral response functions and calibration of the visible channels. This could lead to different SAL values for different instruments. E.g. during the CM SAF CDOP-1 phase the SAL values derived from MSG and NOAA polar orbiting satellites showed a difference of several per cent albeit identical retrieval methods were applied. Further, SAL can only be retrieved accurately in

cloud free skies. This is particular of importance for regions with changing snow cover due to fresh snowfall or melting periods as this are the main driver for huge short-term changes in SAL. Respective changes can not be detected provided there are long lasting clouds. Snow is associated with high surface albedos and it's miss-classification leads to large errors in the retrieved SSI values. The uncertainties of SAL retrievals are discussed in more detail in Shuai et al. (2020); Carrer et al. (2010, 2018) and Fell et al. (2021). They hamper an accurate retrieval of $COD$, $r_{eff}$ and aerosols. Surface albedo data is available from LSA-

SAF COPERNICUS (Geiger et al., 2008; Carrer et al., 2010, 2018) , CM-SAF (Schulz et al., 2008), COPERNICUS NOAA / NESDIS (Wang et al., 2013), GEWEX (GEWEX-Quarterly) and several other sources. To the knowledge of the authors the best data source for snow coverage is the IMS snow mask Helfrich et al. (2018), however this mask is provided only in a daily resolution. Again, data from modern NWP models might be a good alternative as satellite information is assimilated. This option should be investigated in more detail.

### 7.3.4 Aerosols

Aerosol information has the highest uncertainty among the atmospheric clear sky parameter. This results from the gap of a dense ground based network and the limitations of satellite retrieval methods. The Aeronet network provide high quality data, but the station density is too low to gain an accurate gridded data set. Satellite based AOD can not fill this gap as the retrieval of aerosols (AOD and type) is ill-posed. The reflection signal is weak compared to that of SAL and clouds. Therefore,

quite accurate cloud screening and accurate spectral resolved information of the surface reflection is needed. However, optical thin cirrus clouds are hard to detect and the uncertainty of SAL retrieval is in the same order of magnitude as the signal from aerosols for low to medium aerosol loads (RTM study CM SAF, not published). Further, for a given AOD the reflection depends also on the composition and the shape of the aerosols (aerosol type), and visa versa. Absorbing aerosols (e.g. urban aerosols) tend to lead to a reduction of the reflection for increasing AOD whereas scattering aerosols (desert, rural, maritime...

) lead to a significant increase in the reflection (Müller et al., 2016). Unfortunately, the possibilities for the accurate retrieval of aerosol composition and shape from operational satellites is very limited. ADMs/BRDFs are needed for the retrieval as well. Although, significant progress has been made using MODIS data (Lipponen et al., 2018), accurate estimation of aerosol





information is still a great challenge. MODIS is equiped with a blue (dark) channel which favours the retrieval of aerosols as SAL is relativ dark over vegetated surfaces in this wavelength region. However, accurate calibration is a serious challenge

in the blue channels due to the exposure with UV, leading to ageing of the optical devices (e.g. the diffuser). In summary, satellite based aerosol data comes with serious uncertainties and the aerosol information is only provided for cloud free skies, yet, the AOD depends for hygroscopic aerosols on the relative humidity (Hess et al., 1998) and thus changes with cloudiness. Based on the validation results and RTM studies of CM SAF, the authors are not aware of any aerosol data set that captures the temporal and spatial variation of AOD with a higher accuracy of approximately $+/- 0.1$. This might explain why amazingly

good accuracy of SSI can be gained over Europe with a fixed visibility of about 23 km (Ineichen et al., 2009). Within this scope it is important to consider that the variation in AOD is on average relative low in many regions compared to its uncertainties, and that 23 km visibility is a reasonable climatological assumption (see Aeronet measurements and Müller et al. (2016)). Thus, accurate information of the spatial and temporal aerosol distribution are needed to beat on average climatological values or fixed visibilities. As a result of the disadvantages of the individual data sources (i.e. ground based measurements, satellite

based data sets and chemical transport models) a reasonable way forward might be the use of aerosol information from data assimilation, the combination of numerical modelling, satellites and ground based measurements. At least for the CALSAT methods it has been shown that this approach has a great potential to provide better aerosol data sets (Mueller and Träger-Chatterjee, 2014; Müller et al., 2016). Therefore, MACC reanalysis data (Inness et al., 2013) was used for the generation of the CM SAF SARAH data sets e.g. (Müller et al., 2015; Mueller and Trentmann, 2015; Pfeifroth et al., 2019b). The MACC

procedure consists of a forward model for aerosol composition and dynamics (Morcrette et al., 2009) and a data assimilation procedure (Benedetti et al., 2009). The data set is available at MACC-Weblink and has been used to assess the radiative forcing by aerosols (Bellouin et al., 2013). It is recommended to consider this data set for aerosol information within the retrieval of SSI. Yet, for CAL methods, a specific feature enters the door. High fluctuations of desert aerosols (and familiar types) relative to the background, e.g. as induced by desert storms, are interpreted as clouds, because the reflection is significantly increased

compared to clear sky reflection. This has to be considered and a modification of the used aerosol information might be of benefit for CAL methods. An respective approach is presented and discussed in more detail in Müller et al. (2016).

### 7.3.5 Turbidity

Until the Helioat-3 project Linke turbidity was usually used as input for the clear sky model applied within Heliosat methods (Ineichen and Perez, 2002; Beyer et al., 1996; Cano et al., 1986; Rigollier et al., 2004; Perez et al., 2001) instead of separate

aerosols and water vapour information. However, the use of Linke turbidity has a serious disadvantage. For the accurate calculation of the ratio between diffuse and direct irradiance, information of the aerosol type and the aerosol optical depth are required (Mueller et al., 2004). An exact ratio of direct and diffuse radiation is essential for an accurate estimation of the irradiation on tilted surfaces that are typical for PV systems. The use of turbidity was therefore in contradiction to the aims of the Heliosat-3 project to support the solar energy community in its efforts for increasing the efficiency and cost-effectiveness of

solar energy systems and improving the acceptability of renewables. Therefore, RTM based methods were developed (Mueller et al., 2004) in the Heliosat-3 project, which are able to consider the effect of AOD and aerosols type, water vapour as well as





ozone separately. These concept and input information were already used in the climate community (Gupta et al. (1999) and references therein). Thus, nowadays a clear sky model or parameterisation should be able to use the full set of information of the dominant atmospheric variables, aerosol optical depth and aerosol type, water vapour content and ozone. See Mueller
et al. (2004) for further discussions and details. However, as a result of the uncertainties in aerosol information the use of well calibrated ground based turbidity data lead still to remarkable good results.

### 7.3.6    Satellite

The new satellite generations (GOES, HIMAWARI, Meteosat Third Generation) offer a higher spatial and temporal resolution and are equiped with more spectral channels. These features enables a better distinction between clouds and snow. Overall,
it can be expected that the new satellite generations will increase the accuracy of solar surface radiation data. A particular new feature for Remote Sensing in Europe and Africa is the blue channel available with Meteosat Third Generation. In this wavelength spectrum most land surface types can be assumed to be dark surfaces. This in turn enables a more accurate retrieval of aerosol from geostationary satellites, because, their is a better option to estimate the aerosol type and the absolute errors induced by surface albedo are minimised. Further details of the enhanced features can be found on the MTG web page (MTG-
Weblink).

## 8    Conclusions

Satellite-based remote sensing of solar surface irradiance (SSI) is a key component in monitoring and predicting the state of the earth's atmosphere and is also of great importance for the planing, monitoring and operation of solar energy systems.

The effective cloud albedo $CAL$, also reffered to as cloud index ore effective cloud fraction, can be retrieved directly from
the radiances observed by the weather satellites without the need of any modelling or external informaion (Mueller et al., 2011). The respective CALSAT approach was initially developed by Möser and Raschke (1984); Cano et al. (1986) and Diekmann et al. (1988). The basic relationship between observed radiances and the effectice cloud albedo (equation 8) is still the basis for state of the art retrievals and for estimating SSI with an accuracy close to that of well-maintained ground-based stations (Dürr and Zelenka, 2009; Posselt et al., 2011b; Mueller et al., 2012a; Müller et al., 2015; Castelli et al., 2014; Pfeifroth et al., 2019b).
There are good reasons why the CALSAT approach is still very prominent and widely used. It is based on the physical law of energy conservation and takes full benefit of the satellite observations for the retrieval of the cloud transmission. There is no need for any ADM simulation or external information of the surface albedo or to asssume that clouds are plane-parall. Only the observed reflections are needed to define the effective cloud albedo and thus the cloud transmission. This reduces the uncertainty induced by 3rd party information or assumptions needed to model fluxes, which in turn are necessary for the
indirect methods.

Therefore, the authors do not see any advantage in using indirect ways, e.g. to retrieve $COD$ and $r_{eff}$ for the estimation of the cloud transmission. For the retrieval of $COD$ and $r_{eff}$ at least two visible channels are needed, which are not available for for Meteosat First Generation satellites as the MVIRI instrument is equipped with only on visible channel. This is a serious





limitation for the generation of climate data records. $CAL$ can be derived from all Meteosat satellite generations and can be
easily adapted to other satellite systems. However, CALSAT methods provide only information about the cloud transmission,
but not about the micro-physical cloud properties. If micro-physical quantities of clouds are needed or of interest then retrieval
methods based on $COD$ and $r_{eff}$ are the preliminary choice. Depending on the area of application, it can then also be useful
to use $COD$ and $r_{eff}$ for the determination of the solar surface radiation in order to obtain a consistent data set.

Concepts developed within HELIOSAT-3 (Mueller et al., 2004) went eventually in the development of the methods MAGIC
(Mueller et al., 2009, 2011; Posselt et al., 2011b) and SPECMAGIC (Mueller et al., 2012a). In these methods the cloud
transmission derived with the CALSAT approach is coupled with RTM-LUT based clear sky model. In this way, the advantages
of the CALSAT approach can be fully exploited and at the same time the complete clear sky information can be used instead of
only turbidity. The concept of eigenvector hybrid LUTs (Mueller et al., 2009) enable a significant reduction of the needed RTM
calculations and is recommended therefore. SPECMAGIC was used for the generation of the first CDR of spectral resolved
irradiance and is the basis of all CM SAF SARAH data sets.

Of course, satellite retrievals has also limitations. That is the reason why during the last decades several extensions has been
done to improve the CALSAT based methods, including a specific and powerful method for mountains, the Heliomont method
(Dürr and Zelenka, 2009; Stöckli, 2013). The improvements of the method cover parallax correction, slant geometry correction,
snow-cloud differentiation, calibration and correction of shadowing effects. Although, significantly improvements have been
achieved more studies and works are needed to gain further improvements. However, it is likely that significant increase of
the SSI quality can be gained from an improved accuracy of information in cloud-free skies, in particular concerning aerosol
optical depth and type (Müller et al., 2016). For the clear sky input, due to the extension and further development of data
assimilation, ECMWF data might be the best choice. Finally, the upcoming satellite generations, having a higher resolution
and more spectral channels, will likely contribute to the further improvement of atmospheric input information and retrieval
schemes.

*Code and data availability.* The near real time and forecast data is available at opendata.dwd.de/weather/satellite/radiation/, the CDRs at
wui.cmsaf.eu, the code is available on request (GPL license). Please contact Richard.Mueller@dwd.de



**Meaning of eigenvector approach**

A process is called independent of other atmospheric processes if for a given deviation of an atmospheric variable (here water
vapour for example) a unique scalar $t_{H_2O}$ can be defined which fulfils the following equation.

$$RTM_{\delta H_2O} I_0 = t_{H_2O} \cdot I_0 \tag{1}$$

$I_0$ is the extraterrestrial irradiance and $RTM_{\delta H_2O}$ is an operator, describing the effect of deviations in water vapour on $I_0$.
For every $RTM_{\delta H_2O}$ a unique $t_{H_2O}$ exists which depends only on the amount of water vapour and on no other atmospheric
variable. The value of t describes the atmospheric transmission. The atmospheric transmission of the operator $RTM_{\delta H_2O}$
depends only on the amount of water vapour. $t_{H_2O}$ can therefore be interpreted as eigen-value of the operator $RTM_{\delta H_2O}$ and
$I_o$ as "eigenvector".

For aerosols this is not the case, because no eigenvalue exists. Hence,

$$RTM_{\delta AOD} I_0 \neq t_{aod} \cdot I_0 \tag{2}$$

For equal $\delta AOD$ not a unique but different $t_{aod}$ values exist, as $t$ depends on the value of aod (aerosol optical depth), ssa
(single scattering albedo) and gg (asymmetry factor). The transmission for a given aod depends also on the values for ssa and
gg.

For the surface albedo, Equation 1 is also valid not for the extraterrestrial irradiance but the surface irradiance:

$$\delta SAL I_{surf} = t_{H_2O} \cdot I_{surf} \tag{3}$$

*Author contributions.* Richard Müller wrote the manuscript. Uwe Pfeifroth contributed to the writing.

*Competing interests.* We declare that we have no competing interest at present.

*Acknowledgements.* We thank the CM SAF team for the support of the development and generation of the solar surface irradiance data sets
and EUMETSAT for the funding of CM SAF. Further we thank our families for their support, which enabled us to make our studies.



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
