# Peer review of "Remote Sensing of solar surface radiation - A reflection of concepts, applications and input data based on experience with the effective cloud albedo"

_Atmospheric Measurement Techniques, 2021_

## Author Comment (AC1)

**Authors' Responses to Reviewer 1:**

The authors provide an extensive overview about the CM-SAF CAL approach for retrieving the SSI from satellite measurements. In the introduction, the authors also mentioned other methods to derive the SSI and referred to many publications. Then the authors mainly explained the CM-SAF SSI algorithm with new development and ideas. It is amazing that the authors could include so many topics in one paper. The structure of the paper could be improved. Some subsections seem not at the right place. I think the paper fits the scopes of AMT and can be published after some corrections.

Specific comments

**Line 57,**

'500 nm' should be '500 m'

→ Thanks, corrected.

**line 75**

'The value of satellite data is further increased due to the automation of ground based networks. '

Could you add an explanation for this sentence?

→ Thanks: The following sentence is added to the manuscript. „E.g. at DWD the satellite-based direct irradiance is used to derive raster data of sunshine duration as a replacement of the former Campbell–Stokes recorders .

**Paragraph from line 114.**

The authors reviewed some SSI data sources. The KNMI CPP -SICCS data set is also available online. Actually the CM- SAF cloud properties are retrieved using the CPP algorithm and SICCS SSI products derived from the cloud properties. https://msgcpp.knmi.nl/.

→ Thanks, we added the following information to the manuscript: „*Solar radiation data are also available online from KNMI ({\it msgcpp.knmi.nl}). The data is based on the Cloud Physical Properties (CPP) algorithm, which is being developed at KNMI to derive cloud, precipitation and radiation products from satellite instruments (e.g. SEVIRI). The development was partly funded*

*by Eumetsat within the scope of the CM SAF activities. Open Geospatial Consortium (OGC) services are used to offer the near real time products."*

todo

\# In the end of section 1, I think it is better to include a few sentences to present an outline of the rest of the paper.

→ Thanks: A few sentences for the outline were added to the manuscript. Line 131-136

\# Line 145 please correct the typo 'emmitted'

→ Thanks, done.

\# line 184 'many RTMs must be performed …'

Do you mean 'many RTM calculations must be performed …'?

→ Yes, thanks, corrected.

\# line 186 , '… recalcuation , whiis necessary from time …'

please correct the typo.

→ Thanks, corrected

\# Line 196 '… using the DISORT solver (Stammes et al., 1988) …'

Please correct the author name, it should be 'Stamnes'.

→ Thanks, corrected.

\# Line 197 ' … resulting from a adaptation of the Skartveit et al. (1998) …'

change 'a' to 'an'

→ Thanks, done.

\#Eq. 5 looks the same as Eq. 3, Is it needed here?

→ Thanks, indeed, they are identical, thus we replaced the equation by a reference to eqzuation 2

\# line 210 , ' ... hence almost all of the UV-A radiation reaching the top of …'

I think 'UV-A' should be changed to 'UV-B' according to the content of the sentence.

→ Thanks we rephrased the sentence: „Almost all UV-B radiation that enters the atmosphere is absorbed"

**line 218 ' In contrast, cloud droplets and aerosols are leading to Mie scattering, …'**

It is not accurate to include aerosols here. Because some aerosols particles are not spherical, Mie scattering is not a good approximation for aerosol scattering.

Please rewrite the sentence.

→ Thanks, we agree, aerosols is to general and misleading in this context. We rephrased the sentence as follows: *„In contrast, cloud droplets and spherical aerosols are leading to Mie scattering, …"*

**Line 223 ' The great majority of air molecules (N2 , O2 , CO2 , methane, noble and inert gases) are well mixed and uniformly distributed and do not affect the spatial and temporal distribution of solar surface irradiance. '**

Please rewrite this sentence. CO2 and methane could be well mixed vertically in the atmosphere but there are spatial and temporal variations. Of course the variations are small but they are not in the came category as O2 and N2. The sentence in line 223 could be misleading.

→ Thanks: We rephrased it to: *„The great majority of air molecules ($N_2$, $O_2$, noble and inert gases) are well mixed and uniformly distributed and do not affect the spatial and temporal variation of solar surface irradiance. Even the rather small fluctuations of methane and $CO_2$ have no significant effect on SSI variation."*

**Line 258 ' Is the change in the fluxes induced by a different viewing geometry for the same SAL or by the change in SAL induced by SZA, change in vegetation, different pixel size, calibration issues (ageing of channels, change of spectral response function), change of satellite instruments, and others. '**

This sentence is not easy to read. Please rewrite it.

→ Thanks, we rephrased it to *„The relation between observed radiances and simulated fluxes depends not only on the viewing geometry but also on landuse (SAL SZA dependency), pixel size, calibration issues (ageing of channels, change of spectral response function) and other effects."*

**Line 293**

'In brief, using the indirect approach means to be further away from the observations, to introduce further error sources by weak assumptions and to use simulations instead of observations directly.'

I think this statement is too negative about the indirect approach. I would write it differently.

 -> Thanks, we agree. We deleted the word weak and the ill-posed phrase rephrased the  paragraph to  *„In brief, using the indirect approach means to be further away from the observations, as additional assumptions and simulations are needed in contrast to the direct path, which is discussed in the next section.“*

**line 367**

' … cloud transmission ist quite …' correct the typo

→ Thanks, done.

section 6 'Forecasting and seamless prediction'

I do not see the authors mentioned anything about the seamless prediction. Could the authors add a short paragraph?

→ Thanks: A detailed discussion of seamless prediction would be out of the scope of the manuscript and seamless prediction was therefore deleted from the section title, but for completeness we added the following sentence to the manuscript:  *„However, cloud motion vector methods have the disatvantage that they can not capture convection or dissipiation (change in intensity). Although NWP has also great difficulties in this area they include at least physical parameterisations to deal with the phenomena.  That is the reason why after a couple of hours NWP models outperform NWC and a combination of both methods is needed in order to gain the optimal accuracy for every time step.“*

*A more detailed discussion of seamless prediction would be out of the scope of the manuscript*

**line 541.**

Please add some comments about the ECMWF snow/ice forecast product.

 → Please apologize, but we did not capture what is meant here. In the manuscript NIC IMS snow mask is mentioned and referred. We think a discussion of the ECMWF snow/ice forecast product might be out of the scope.

**Section 7.2.1 around line 600. The authors commented that the the error reduction by implementing the parallax correction is marginal. I think the authors look at the monthly mean or in a large area.**

For a specific location having some small clouds, the parallax correction is important. Perhaps the authors could add some discussions on this case. Perhaps there are not much corrections for the SSI values but the SSI values have to be assigned at the right pixels.

→ The comment is based on the communication with Reto Stöckli from MeteoSwiss. The error reduction was marginal compared to the other „mountain" specific sources. Of course, the parallax correction could be more significant in homogenneous terrain.  We added a sentence to clarify this. *„Especially since the parallax correction could be more significant in homogeneous terrain."*

**7.2.3 Deep learning - …**

I think this deep learning section does not belong to 7.2. It could be a new subsection, 7.3 or 7.4.

→ Thanks, section 7.2 was incorrectly a section but should be a subsection. Has been modified accordingly.

**7.3.1 Ozone**

The discussion of impact of ozone is only on the broadband SSI. Since the authors also discussed the spectral resolved irradiance, the readers may want to know the impact of ozone on the SSI in the UV wavelengths.

→ Thanks: We added the following sentence to the manuscript: "*However, the absorption effect of ozone in the UV is very strong and therefore accurate information about the ozone concentration is required for this spectral range.*"

**7.3.4. Aerosols**

Could you comment on the CAMS aerosol forecast product?

→ Thanks: We added the following text to the manuscript. *„The research and developments of the MACC projects went into the Copernicus Atmosphere Monitoring Servicce (CAMS)  \cite{Innes_19}, which was implemented by ECMWF as part of the Copernicus Programme. Thus, CAMS is some kind of successor of MACC and a very valuable source for aerosol information. CAMS provides reanalysis as well as forecast data.*

**Line 774 'The effective cloud albedo CAL, also reffered to as cloud index ore effective cloud fraction …'**

I think the OMI SSI product using the effective cloud fraction and its references can be referred to . (https://www.temis.nl/ssi/).

→ Thanks, this is now mentioned and cited at the end of section 4

Please correct the typos.

→ Done

---

## Author Comment (AC2)

**Authors' Responses to Reviewer 2:**

General Comments:

Muller and Pfeifroth present a review paper summarizing the physics and techniques that go into satellite based retrievals of solar surface irradiance data (SSI). The authors give a nice overview of the subject with extensive references. With the ever increasing need for accurate SSI data this paper should serve as a useful reference in the field.

The authors also present a good argument for the well-established technique of using the observed effective cloud albedo (CAL) for the retrieval of the cloud transmission as opposed to the more indirect method of determining cloud transmission from the derived microphysical parameters of the cloud (cloud optical depth and effective radii) and RTM.

The English grammar could use some work, but my suggested edits below should help.

Overall, this paper addresses a relevant scientific topic within the scope of AMT. I recommend acceptance after minor revisions.

Specific Comments:

Abstract: The abstract is concise and provides a good summary of the paper.

→ Thanks.

Section 1, Introduction: The authors do a good job in the introduction of motivating the need for accurate SSI data, why satellite observations are the most useful for getting SSI, and in explaining the current satellite data sources.

→ Thanks

Section 2: The authors give a good overview of the basics of the methods and physics that go into deriving SSI from satellite data.

→ Thanks

Section 3 and 4: The authors do a good job of describing the indirect and direct path methods.

→ Thanks

In Section 1, Lines 99-101: A reference, or web link, for GEWEX should be given

→ Thanks. We added the GEWEX web link.

In Section 1, Lines 109-113:  A reference, or web link, for CERES should be give.

→ Thanks. We added the CERES web link.

In Section 2.1, Equation 4, dcor should be explicitly defined

→ Thanks, we added „$d_{cor}$ is a correction factor for varying distance between the Earth and the Sun"

Line 216:  The wavelength dependence for Rayleigh scattering should be lambda to the negative 4

→ Thanks, corrected.

Section 7.1, Lines 450-459:  These lines seem out of place in this paper. They are more of a philosophical lecture that comes across as condescending to the reader.  I think these lines should be deleted.  Just get right into the Heliosat project and the CALSAT method, why the cloud microphysical method was suggested, etc.

→ Yes, we agree it is a bit philosophical, we deleted the paragraph and replaced it by.

„A central paradigm of science is the goal to make things as simple as possible and as complex as necessary and to benefit from the knowledge gained. In this context, we we would like to briefly review the lessons learned within the Heliosat-3 project."

Section 7.1.1, Lines 487-488:  The text reads as if the HRV channel is only available over Europe, and only applied over the European mountains, which I don't think is the case, is it?  These 2 sentences should perhaps be more generalized to indicate the HRV channel improves things not only over the European mountains, but over all mountainous terrain.

→ Thanks, we apologize the misleading phrase, the RSS service is only availble over Europe. HRV covers parts of Africa.  We added „ and large parts of Africa"

Section 7.2, Lines 570-578:  I don't think there's a need to actually list the C-code here.  My opinion is that it is too much detail for this type of paper and could be taken out.

→ Thanks, we moved the C-code lines to the appendix.

Technical Corrections:

Here are suggested grammatical corrections.  They are minor corrections, but there are a lot of them:

Line 40:  Change to "and 9.3% in Germany in 2020"

→ Thanks. done.

Line 51:  Remove redundant phrase "flying in low orbit around Earth"

→ Thanks: Removed.

Line 60:  Change 'primary' to 'primarily'.   Change 'conversation' to 'conservation'

→ Thanks, Done

Line 65:  Spell out acronym 'BSRN' the first time used

→ Thanks, Done

Line 80:  Change 'provider' to 'providers'

→ Thanks, Done

Line 85:  Change to "…based solar irradiance: the Climate…"

→ Thanks, Done

Line 88:  Delete 'also'

→ Thanks, Done

Line 109:  Suggest starting new paragraph at "The Clouds and the Earth's …"

→ Thanks, Done

Line 117:  Change to "   PVGIS allows one to visualize…"   Change to "…selected sites.  Different data…"

→ Thanks, Done

Line 118:  Start new sentence…"…sources can be selected.  In addition to …" and deleted 'also' at end of line.

→ Thanks, Done

Line 119:  Change to "…(Hersbach et al., 2020) are available."

→ Thanks, Done

Line 119:  Should 'servide' be 'service'?

→ Yes, corrected

Line 120: Change to "SoDa was commercilised…"

→ Thanks, Done

Line 153: Change to "…constant is somewhat misleading…"

→ Thanks, Done

Line 157: Change to "…the year on the order of …"

→ Thanks, We believe „the year in the order of" is correct.

Line 158: Change to "…leading to a respective…"

→ Thanks, Done

Line 174: Change to "powerful"

→ Thanks, Done

Line 187: Delete 'also'

→ Thanks, Done

Line 187: I don't think it's correct to start a sentence with 'E.g.' I think you should spell it out at the beginning of a sentence, that is "For example, after 4 years of…" The use of 'E.g.' at the beginning of the sentence was done a few times throughout the paper, so all should be changed to 'For example,…"

→ Thanks. Replaced e.g. by for example

Line 194: Change to 'calculations'

→ Thanks: Done

Line 205: Change 'nevertheless' to 'is somewhat'

→ Thanks: Done

Line 207: Change 'modify' to 'to modification of'

→ Thanks: Done

Line 210: Change 'E.g.' to 'For example,'

→ Thanks: Done

Line 212: Change to 'acting as a strong…"

→ Thanks: Done

Line 218: Change 'are leading to 'follow'

→ Thanks: Done

Line 238: Change 'hybrids' to 'hybrid'

→ Thanks: Done

Line 246: Change 'neglected in' to 'neglected to'

→ Thanks: Done

Line 246: Change 'trough' to 'through'

→ Thanks: Done

Line 247: Change 'equals in' to 'equals to'

→ Thanks: Done

Line 263: Is there a reference for Skartveit and Olseth?

→ No, it was a personal communication during the Heliosat-3 project. This info has been added to the mansucript.

Line 263: Add a comma after 'indirect path,'

→ Thanks: Done

Line 288: Add 'to as the prototype'

→ Thanks: Done

Line 298: Change 'induces' to 'induce'. Add 'the use of the direct path'

→ Thanks: Done

Line 351: Change 'has been retrieved' to 'have been retrieved'

→ Thanks: Done

Line 354 and 355: The reference 'Kulesza, 2021' is repeated

→ Thanks: Is fixed.

Line 367: Change 'ist' to 'is'

→ Thanks: Done

Line 368: Delete 'used and for' to 'used for'

→ Thanks: Done

Line 396: Change 'a adaption' to 'an adaptation'

→ Thanks: Done

Line 407: Change 'larger' to 'longer'

→ Thanks: Done

Line 424:  Change 'E.g.' to 'For example, '

→ Thanks: Done

Line 435:  Change 'met' to 'meet'

→ Thanks: Done

Line 438:  Change 'Further, a running' to 'Further, running'

→ Thanks: Done

Line 446-447:  These lines seem out of place and redundant here at the end of this section.  Seems like they should go at the beginning of the section where you are motivating the need to forecast SSI

→ Thanks the sentence has been moved to the introduction of the section.

Line 470:  Change to 'approach as not been replaced by'

→ Thanks: Done

Line 472:  Change misspelling to 'conservation'

→ Thanks: Done

Line 473:  Change to 'In these terms'

→ Thanks: Done

Live 482:  Change to 'possible improvements to these retrievals'

→ Thanks: Done

Lines 484-485:  Change to 'is relatively coarse, approximately 3-5 km over the European mountains.'

→ Thanks: Done

Line 488:  Change to 'which means about', that is, remove 'of'

→ Thanks: Done

Line 493:  Change 'digitally' to 'digital'

→ Thanks: Done

Line 503:  Change 'pixel' to 'pixels'

→ Thanks: Done

Line 509:  Change 'rho' to the actual Greek letter

→ Thanks: Done

Line 539:  Add 'satellite information lines including…'

→ Thanks: We changed satellite information to satellite instruments.

Line 540:  Change to 'However, melting periods of snow are still sometimes…'

→ Thanks: Done

Line 587:  Change to 'independent of cloud height'

→ Thanks: Done

Line 589:  Change 'E.g.' to 'For example, '

→ Thanks: Done

Line 610:  Change to 'Based on those results'

→ Thanks: Changes it to ...on these...

Line 622:  Add acronym for neural networks here, that is 'neural networks (NN)' then you don't have to keep redefining it throughout the first paragraph on pg. 24

→ Thanks: Done

Line 623:  Change to 'This also includes the estimation…'

→ Thanks: Done

Line 625:  Change to 'neural networks is their black box character'

→ Thanks: Done

Lines 652-653:  Change to 'However, climatologies can also be used in…'

→ Thanks: Done

Line 661:  Remove 'European Medium Range Weather Forecast' since you already defined ECMWF earlier

→ Thanks: Done

Line 665:  Remove 'of' at end of sentence, that is, 'SAL is about'

→ Thanks: Done

Line 668:  Change 'E. g.' to 'For example,'

→ Thanks: Done

Line 670:  Change to 'This is of particular importance for regions…'

→ Thanks: Done

Line 671: Change to 'as these are the main drivers for huge…'

→ Thanks: Done

Line 672: Change to 'and its miss-classification'

→ Thanks: Done

Line 681: Change to 'parameters'

→ Thanks: Done

Line 694: Change to 'relatively dark'

→ Thanks: Done

Line 697: Change to 'yet, for hygroscopic aerosols the AOD depends on the relative humidity…'

→ Thanks: Done

Line 699: Change to 'with an accuracy higher than approximately'

→ Thanks: Done

Line 701: Change to 'relatively'

→ Thanks: Done

Line 707: Change to 'has great potential'

→ Thanks: Done

Line 718: Change misspelling to 'Heliosat'

→ Thanks: Done

Line 727: Change to 'concepts'

→ Thanks: Done

Line 734: Change to 'enable'

→ Thanks: Done

Line 744: Change to 'index or'

→ Thanks: Done

Line 752: Change to 'plane-parallel'

→ Thanks: Done

Line 764: Change to 'eventually into the development'

→ Thanks: Done

Line 769:  Change to 'and is therefore recommended'

→ Thanks: Done

Line 771:  Change to 'retrievals also have limitations'

→ Thanks: Done

Line 771:  Change to 'extensions have'

→ Thanks: Done

Line 774:  Change 'significantly' to 'significant'

→ Thanks: Done

Line 775:  Change 'works' to 'work'

→ Thanks: Done

Section 'Meaning of eigenvector approach'.  Shouldn't this be labelled as an Appendix

→ Thanks: It is in the appendix section in the tex file, we added "appendix" to be concise.